


# A risk-based, network analysis of distributed in-stream leaky barriers for flood risk management

Barry Hankin[1,2], Ian Hewitt[3], Graham Sander[4], Federico Danieli[3], Giuseppe Formetta[5], Alissa Kamilova[3], Ann Kretzschmar[2], Kris Kiradjiev[3], Clint Wong[3], Sam Pegler[6], Rob Lamb[7,1]

[1]Lancaster Environment Centre, Lancaster University, Lancaster, UK
[2]JBA Consulting, Skipton, UK
[3]Mathematical Institute, Oxford University, Oxford, UK
[4]School of Civil and Building Engineering, Loughborough University, UK
[5]Department of Civil, Environmental and Mechanical Engineering, University of Trento, Italy
[6]School of Mathematics, University of Leeds, UK
[7]Director, JBA Trust, Skipton, UK

*Correspondence to*: Barry Hankin (b.hankin@lancaster.ac.uk)

**Abstract.** We develop a network-based model of a catchment basin that incorporates the possibility of small-scale, in-channel, leaky barriers as flood attenuation features, on each of the edges of the network. The model can be used to understand effective risk reduction strategies considering the whole-system performance; here we focus on identifying network dam placements promoting effective dynamic utilisation of storage, and placements that also reduce risk of breach or cascade failure of dams during high flows. We first demonstrate the model using idealised networks and explore risk of cascade failure using probabilistic barrier-fragility assumptions. The investigation highlights the need for robust design of nature-based measures, to avoid inadvertent exposure of communities to a flood risk, and we conclude that the principle of building the leaky-barriers on the upstream tributaries is generally less risky than building on the main trunk, although this may depend on the network structure specific to the catchment under study. The efficient scheme permits rapid assessment of performance of dams placed in different locations in real networks, demonstrated in application to a real system of leaky barriers built in Penny Gill, a stream in the West Cumbria region of Britain and which leads to further design advice.

## 1 Introduction

The concept of "green infrastructure" is embedded within environmental policy in Europe (European Commission, 2007, 2013a,b, EEA, 2015) and the UK (Defra, 2019) as a strategic approach involving the design and management of networks of natural and semi-natural environmental features to deliver a wide range of ecosystem services. Echoing this approach, projects around the world have been blending natural and engineering approaches to deliver multiple social and environmental benefits (WWF, 2016 , Bridges et al., 2018). In Flood and Coastal Risk Management (FCRM) there has been a growing interest in so-called `nature-based' measures, including small-scale, distributed storage features, tree planting and soil structure improvement to prevent fast overland flow. These measures have collectively become known as Natural Flood Management (NFM) in the UK (see Dadson et al., 2017 and Lane, 2017), or Working With Natural Processes (WWNP) after the Pitt Review of the UK 2007 summer floods (Pitt, 2008), a term adopted in the recent UK Evidence Directory (Burgess-Gamble et al., 2018). Internationally they have also been termed "nature-based approaches" or "engineering with nature" (Bridges et al., 2018).

One such nature-based measure is to encourage in-channel flood attenuation (e.g. see Metcalfe et al., 2017), using small dams or barriers, usually made from wood (Figure 1). These barriers, which are often deliberately built to be permeable (and sometimes called "leaky barriers"), allow low flows to pass under or through, but hold back high flows, providing temporary water storage analogous to beaver dams. It is hoped that a large collection of such features deployed in a catchment may hold back enough flood water (in-channel or on the floodplain) to mitigate flood risk downstream (Figure 2(a)). In the UK, use of


leaky barriers has been incentivised under the current environmental stewardship grants across England and Wales[1]. However, whilst the effectiveness of systems of runoff attenuation features and leaky barriers has been investigated recently (e.g. Metcalfe et al., 2017 and Addy and Wilkinson, 2019), these studies do not consider performance failure, and there remains much trial-and-error installation of different designs which could be improved upon for more efficient risk-reduction

5    strategies at the large scale.

There have been many attempts at representing the effects of leaky-barriers on flow, with methods ranging from increasing roughness in 1d models to full 3d representation, but relatively few have been able to test the accuracy of the physical representation (see Addy and Wilkinson, 2019). The NERC project, Q-NFM[2] has developed a set of small, accurately monitored 'micro-catchments' in Cumbria to attempt to quantify the effect of different nature-based interventions. One of

10   the micro-catchments is Penny Gill, drains to a small community at risk on the West coast of Cumbria. This stream has had ten robustly constructed leaky barriers (Figure 1 shows two examples from Penny Gill) that are in sequence on the main stem of the stream. The larger, lower eight of these structures were surveyed and have been modelled to try and understand the attenuation and storage during times of flooding, and the system of leaky barriers is used as real-life network to see if the network model presented here can help with design-guidance and effective deployment strategies.

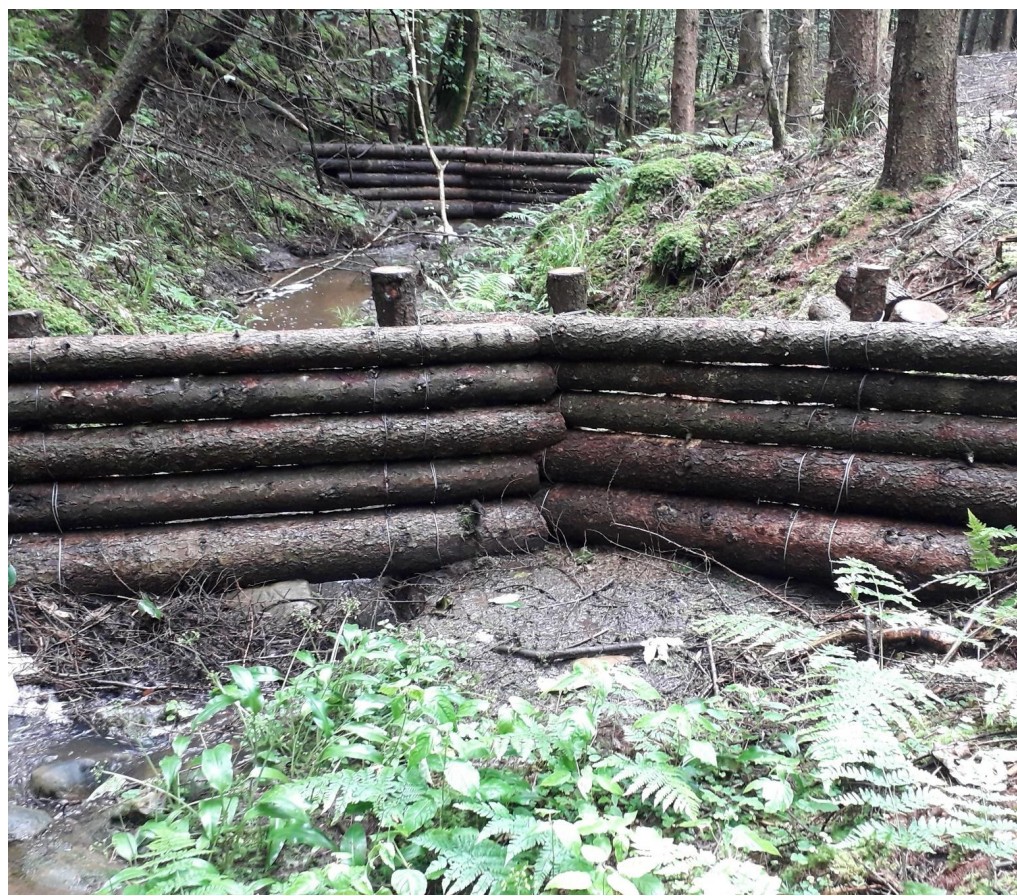

**Figure 1: Leaky barriers in sequence in Penny Gill, Cumbria (B. Hankin)**

---

[1] https://www.gov.uk/countryside-stewardship-grants/rp33-large-leaky-woody-dams
[2] https://www.lancaster.ac.uk/lec/sites/qnfm/





Significant research questions remain about whether "many small interventions (each creating local benefits) [will] combine to create large benefits at large scale" (Dadson et al., 2017) and whether "the lack of demonstrable effect at large scale is because noticeable flood mitigation could not be achieved in a large catchment, or because a sufficiently large-scale set of interventions have not yet been implemented". Meanwhile, in the UK at least, the government's approach is to see working

with natural processes as complementary to conventional, engineered flood risk management measures. In England, this is reflected in the latest long-term investment planning scenarios published by the Environment Agency (2019), whilst noting uncertainty about the effectiveness of NFM to manage large floods and large catchments.

If this complementarity is to be realised in practice, we believe there is a pressing need to integrate NFM more tightly within

the cost, benefit and risk assessment frameworks that apply to "conventional" flood management. This means that we want to understand NFM features as systems of assets, and to assess those systems within a risk-based analysis that considers the whole-system performance in terms of risk reduction. A risk-based analysis of NFM asset systems should take account of both the reliability of the assets and their performance as a whole system under different plausible hazard or loading scenarios. One vital lesson from conventional flood management is that even when flood mitigation measures are in place,

the residual risk cannot be ignored.

Some initial work to test the effectiveness of catchment-wide NFM under a range of spatially distributed extreme rainfalls has been reported by Hankin et al. [2017a], but without consideration of the reliability of the underlying NFM assets. Here, we focus instead on the resilience of a network of NFM features as an asset system. To do this, we develop a simple

network-based model of a river catchment that incorporates the possibility of leaky barriers being installed on each edge of the network, similar to the approach taken by Metcalfe et al. [2017]. We wish to understand the impact of different spatial configurations of the leaky barriers, taking into consideration three possible performance issues. These are:

1) under-utilisation of dynamic storage (see Metcalfe et al., 2018), i.e. redundancy in the network of leaky barriers that could be regarded as an inefficient use of resources;

2) undesired synchronisation of flood peaks (see Pattison et al., 2014), where measures intended to slow the flow could result in flood peaks being increased under some scenarios;

3) structural failure and cascade failure of barriers.

In Section 2 we develop a mathematical drainage network model and show how leaky barriers can be incorporated in a form that is simple enough to enable solution of the resulting system of equations, but sufficiently realistic to describe key

hydraulic modes of behaviour. We then apply the equations in Section 3 to study the performance of idealised one- and two-dimensional stream networks subjected to single-peaked and multi-peaked flood events, including the potential for failure of individual or multiple assets. Multi-peaked flood events are a more effective test to the resilience of the system aimed at providing dynamic storage, that can be re-used on consecutive events, and it is this kind of event that often resulted in more severe impacts. We discuss the findings in terms of the risk reduction and the residual risk achieved by the systems of NFM

features under different configurations, and how the idealised cases may help inform analysis of real NFM systems. In Section 4, the model is applied to the real system of leaky barriers in Penny Gill, West Cumbria and conclusions are drawn about more effective designs and placement.

## 1.1 Performance of existing nature-based dams

There are a number of studies documenting the benefits of beaver dams in terms of habitat improvement, peak flow

attenuation and water quality improvements (Puttock et al, 2017, 2018), so it is natural to try and emulate these types of benefits artificially. However, we should also study what happens in nature when things go wrong. Structural failure of natural beaver dams has been reported as occurring frequently by Butler and Malanson [2005], citing numerous cases of dam




failure, that resulted in outburst floods. These floods have reportedly been "*responsible for 13 deaths and numerous injuries, including significant impacts on railway lines*". Engineered NFM measures are likely to be more robust than beaver dams (contingent on maintenance in the longer term), but the relative risks of different configurations, positioning in relation to geometry, slope and proximity to each other, and build-design need a mechanism for appraisal. The intention is to help

design safer and lower risk configurations of NFM, which is seen as a potentially low-cost complement to conventional flood risk management strategies.

Failure of beaver dam structures in the US has been reasonably well documented (Hillman, 1998; Butler and Malanson, 2005), and there have been two records (Tom Nisbet, *Pers. Comm*.) of leaky barrier performance failure at Pickering, UK to the authors' knowledge, after two large flood events. The first flood event in November 2012 resulted in the washout of one

of the larger dams on the main Pickering Beck and a shift to the edge/bank of a second dam below this. These features were relatively tall structures, and located within a straightened section of channel alongside a railway line, with limited floodplain storage. The logs from the failed dam were caught within the downstream reach between that and a third dam downstream. The failed and shifted dam plus one other were found to be deflecting flows into the river bank, causing some local scouring, placing a local railway at risk of undercutting so they were removed (2014) and replaced with five new dams

on a better reach downstream. The second failure event occurred during the UK Boxing Day floods, 2015, where a total of 11 dams were damaged, all involving a shift/deflection in the dam by edge scour or loss/breakage of top logs, rather than a complete washout. These losses all involved the original, more natural design of cross logs used to construct dams in 2010/11, with no wiring used to secure logs in place. All of these have since been replaced using the now favoured semi-engineered design of horizontal stacked logs secured by wiring (a design also used in Penny Gill – see Figure 1).

Additionally, Addy and Wilkinson [2016] report on complete failure of one structure during a 10% Annual Exceedence Probability (AEP) event for 'engineered log jams' that are albeit designed to trap sediment.

Siting, construction and improvements in engineering design are therefore important, and recent research (Dixon and Sear, 2014) shows logs 2.5 times channel width provide 'near functional immobility' - unlikely to be transported in an extreme event. It is design-construction 'rules of thumb' such as this that we seek here on using mathematics to help reduce flood risk

of the whole system, since such rules are more likely to be used pragmatically in the field.

In this paper we explore network issues impacting the three performance issues categorised above, particularly with respect to spatial configuration of leaky barriers in a network that have a probability of failure defined by a fragility curve, an approach commonly used in the systems approach to quantifying flood risk [Hall et al., 2004]. The probability of failure is very difficult to define for the range of constructions that are being implemented – and how this varies with age, decay and

sedimentation is not known So here we attempt to understand what aspects of geometry, slope, proximity are the best trade-offs for a given, reasonable assumption about fragility. We later translate this back the real world example on Penny Gill, Cumbria, and the implications for spacing and siting.

## 2.0 Method

### 2.1 Introduction

We begin by setting up a network model for an arbitrary stream network, breaking the stream up into segments that may each potentially contain leaky barrier designed to attenuate high flows (often referred to generically as runoff attenuation features). Our aim here is to set out a mathematical formulation for the network of features that will enable us to describe and experiment numerically with different configurations of NFM features within a probabilistic analysis. The model is based on a consideration of essential hydraulic principles, with enough simplification to enable solutions to be obtained

quickly for idealised cases. Rules for the storage and discharge (flux) in each segment are prescribed based on the slope, stream cross-section and roughness. Modifications of these rules to account for the effect of a leaky dam are developed. The model amounts to a series of coupled ordinary differential equations (ODEs) that are solved numerically given





Natural Hazards
and Earth System
prescribed runoff inflow. We then explore solutions for some simple networks forced by idealised flood hydrographs, focussing on the response of the discharge at the downstream end of the network. We then examine the response to failure of the dams including cascade failure.

**2.2 Network set-up**

We construct a network model in which segments of a channel (`reaches') are described in a lumped fashion (Figure 2). The primary variables are the average cross-sectional area $A_i$ and discharge $Q_i$, which flows into the next channel segment downstream. The channel segments correspond to nodes of a graph, and the edges that transfer water downstream can be thought of as potential dams (i.e. the positions at which dams might be added). The connections between the channel

segments are described using an adjacency matrix ($a_{ij}$). The $i^{th}$ row of this matrix is all zeros except for in the $j^{th}$ column, where $j$ indexes the node immediately downstream of the $i^{th}$ node. Idealised network structures, with uniform widths and slopes are used, although positions and connections between the channel segments, as well as their lengths, and slopes, might be determined from studying a real drainage network, for example based on a two-dimensional digital elevation model (DEM) such as that used by Metcalfe et al. [2017], or the Penny Gill example discussed further below.

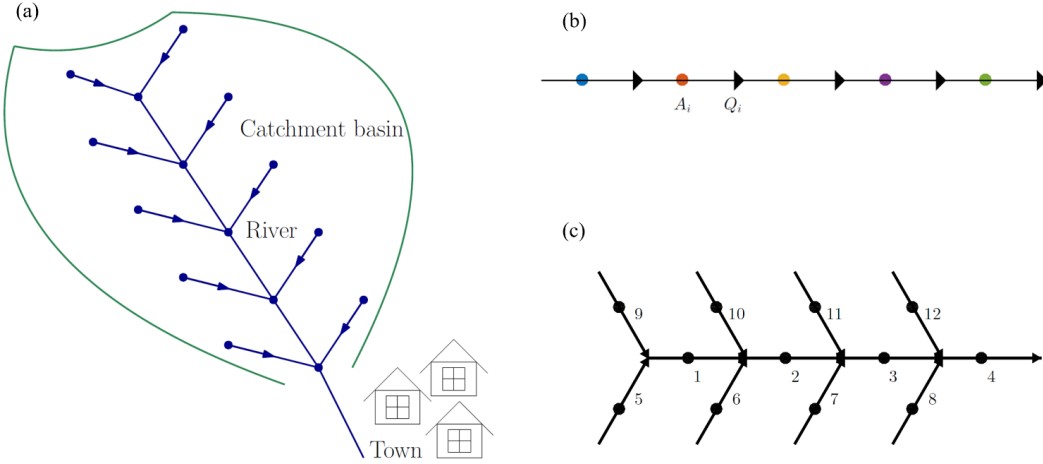

**Figure 2: Schematic (a) and the 1d network (b) and 2d network (c) leaky barrier configurations investigated. Triangles indicate the location of dams which control the discharge, Qi associated with the upstream reach where the average cross section is Ai.**

Taking $l_i$ to represent the length of the channel segments, volume conservation requires:

$$l_i \frac{dA_i}{dt} = \sum_{j=1}^{N} a_{ji} Q_j - Q_i + q_i \qquad (1)$$

for each node ($i = 1 \dots N$). The sum represents the fluxes from the immediately upstream nodes and $q_i$ represents the inflow to each segment from rain/runoff from the surrounding land. It may be more convenient to think of (1) in terms of the water volumes $V_i = A_i l_i$ stored in each channel segment. We assume that the lateral inflows, $q_i(t)$, are prescribed, although in a more complete treatment they might be taken from a two-dimensional model (using the shallow water equations for example), or they might be derived from rainfall data using a filter to represent the time-delay due to subsurface and/or

overland flow.

Given the known slope of each channel segment $S_i$ (which may be related to the bed angle θ by $S_i = \tan(θ)$), we could relate the discharge and cross-sectional area. However, it turns out to be more convenient to express the discharge in terms of the water depth $h_i$ behind the potential dam in each reach. In the case that there is no dam, or when the depth is below the

bottom of the dam, this is simply the average water depth and we can relate this to the cross-sectional area and flow.





The relationship depends on the assumed shape of the channel, and on a parameterisation of turbulent flow. If we assume for simplicity that the channel has a rectangular cross-section with fixed width $w_i$, and use Manning's law, we have

$$A_i = w_i h_i, \qquad Q_i = \frac{w_i^{5/3} h_i^{5/3} S_i^{1/2}}{(w_i + 2h_i)^{2/3} n} \qquad (2)$$

where $n$ is the Manning roughness coefficient and $S_i$ is the slope. Since we can then relate $Q_i$ directly to $A_i$ (by eliminating $h_i$), we can interpret (1) as a set of coupled ordinary differential equations for the $A_i$, forced by the inputs $q_i$. These can be solved numerically using a variety of methods. More generally, when we include dams, we write:

$$A_i = \tilde{A}(h_i; \cdot), \qquad Q_i = \tilde{Q}(h_i; \cdot) \qquad (3)$$

where $\tilde{A}(h_i; \cdot)$ and $\tilde{Q}(h_i; \cdot)$ are known functions, and the extra parameters $(\cdot)$ will describe the dam as well as the cross-section and slope (see below). We also define $\tilde{h}(A; \cdot)$ to be the inverse of $\tilde{A}$ (which is a monotonically increasing function of $h$ and therefore has a well-defined inverse). Thus, we will still have a direct relationship between $Q_i$ and $A_i$.

We take $h$ to represent the height of the water behind the dam. The dam has its bottom at height $b$ above the stream bed, and its top at height $H$. The description of the flow past the dam can then be divided into three modes represented in figure 3, $h < b$ (corresponding to the water level being below the bottom and the dam doing nothing), $b \leq h < H$ (when the dam is operating normally), $h \geq H$ (when the dam is over-spilling).

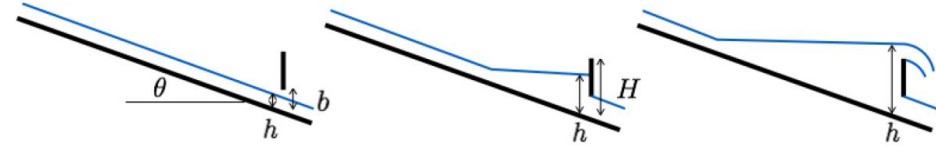

**Figure 3: Flow modes for a leaky barrier.**

For the first mode we use the same Manning relationship as given above to relate $Q$ to $h$. For the second two modes we adopt relationships from hydraulic theory for the flow beneath sluice gates and over weirs (e.g. Munson et al., 2013). When the water depth is part of the way up the face of the dam the flow underneath is given by Bernoulli's equation, to be:

$$wbh \sqrt{\frac{2g}{b+h}} \qquad (4)$$

An empirical correction factor to account for losses is often included in this formula, but we neglect it for simplicity. The flow through the (leaky) dam is assumed to similarly vary with the water depth (due to the hydrostatic pressure), and we write this as:

$$kw(h-b)\sqrt{2gh} \qquad (5)$$

where $k$ should be interpreted as the dam permeability. When the water depth is above the level of the dam, the over-flow is described as for flow over a weir, giving

$$\frac{2\sqrt{2g}}{3} w(h-H)^{3/2} \qquad (6)$$

The leaky flow through the dam is then

$$kw(H-b)\sqrt{2gh} \qquad (7)$$

In summary, therefore,

$$\tilde{Q}(h; w, l, b, H, k, S) = \begin{cases} \dfrac{w^{5/3} h^{5/3} S^{1/2}}{(w+2h)^{2/3} n} & 0 \leq h < b \\[2ex] w\sqrt{2g}\left[\dfrac{bh}{(h+b)^{1/2}} + k(h-b)h^{1/2}\right] & b \leq h < H \\[2ex] w\sqrt{2g}\left[\dfrac{bh}{(h+b)^{1/2}} + k(H-b)h^{1/2} + \dfrac{2}{3}(h-H)^{3/2}\right] & H \leq h \end{cases} \qquad (8)$$





When there is a dam, $A_i$ no longer represents the uniform cross-section of the stream, but rather its average over the length. It is most straightforward to calculate the volume $Al$ in terms of the water depth $h$ for each of the three cases mentioned above. This again depends upon the precise geometry; for the rectangular channel we have

$$\tilde{A}(h; w, l, b, S) = \begin{cases} wh & 0 \le h < b \\ wb + \frac{w(h-b)^2}{2Sl} & b \le h \end{cases} \tag{9}$$

The terms in the second expression here represent the volume of water in the stream up to the depth of the bottom of the dam, plus the volume of water stored in the triangular wedge that forms behind the dam. These relationships are shown in figure 4.

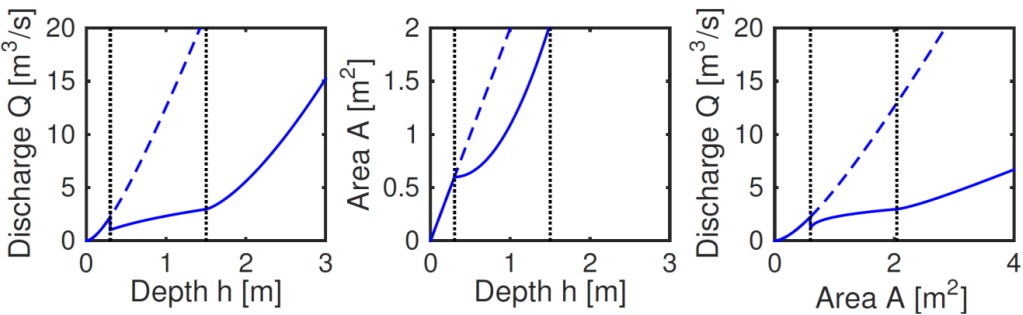

Figure 4: Examples of the relationships between discharge $Q$, cross-sectional area, $A$, and water depth $h$, for rectangular channel
of width $w$=2m, slope $S$=0.01, reach length $l$=100m, and Mannings n of 0.01. Dashed lines show the case of no dam. Solid lines show cases of b= 0.3m, H=1.5m, shown by the vertical dotted lines.

**2.3 Non-dimensional model**

We expect that the flow through the dam will be small compared to that under and over it (the fact that it is allowed to be
leaky makes it easier to construct, but the leakiness between logs is not fundamental to its operation in that there is leaking from underneath the barriers). Thus, for the results presented here, we assume this can be ignored and set the dam permeability coefficient $k$ to zero for simplicity.

We choose scales, denoted by square brackets, such that

$$[Q] = \frac{[w][h]^{5/3}[S]^{1/2}}{n}, \quad [A] = [w][h], \quad [t] = \frac{[l][w][h]}{[Q]} \tag{10}$$

For given typical values of $[Q]$, $[w]$, $[S]$, and $[l]$, these determine the scales $[h]$, $[A]$, and $[t]$. Using typical values for small headwater drainage channels in UK or other humid temperate environments, $[w] = 2m$, $[Q] = 1m^3s^{-1}$, $[S] = 0.01$, and $[n] = 0.01s^{-1}m^{-1/3}$, $[l]$= 100m, we find $[h] = 0.17m$, $[A] = 0.33m^2$, $[t] = 33m^2$.

In non-dimensional form, and assuming negligible dam permeability, these are:

$$\tilde{Q}(h; w, l, b, H, k, S) = \begin{cases} \frac{w^{5/3}h^{5/3}S^{1/2}}{(w+2\alpha h)^{2/3}} & 0 \le h < b \\ w\gamma \left[ \frac{bh}{(h+b)^{1/2}} \right] & b \le h < H \\ w\gamma \left[ \frac{bh}{(h+b)^{1/2}} + \frac{2}{3}(h-H)^{3/2} \right] & H \le h \end{cases} \tag{11}$$

$$\tilde{A}(h; w, l, b, S) = \begin{cases} wh & 0 \le h < b \\ wb + \beta \frac{w(h-b)^2}{2Sl} & b \le h \end{cases} \tag{12}$$

where the dimensionless parameters are

$$\alpha = \frac{[h]}{[w]} \quad \beta = \frac{[h]}{[S][l]} \quad \gamma = \frac{\sqrt{2g}[w][h]^{3/2}}{[Q]} \tag{13}$$




These represent the ratio of depth to width (this is of little importance), the ratio of depth to elevation change across the segments, and the strength of gravity compared to friction –Mannings coefficient is implicit from the relationship for [Q] in (10). For the values given above we find $\alpha \sim 0.08$, $\beta \sim 0.17$, $\gamma \sim 0.6$.

The parameter $\beta$ helps link this mathematical analysis back to the real-world, in that it is related to 'rule of thumb' estimates of backwater length used by hydraulic engineers to understand influence upstream as $0.7*h/S$ (for example see Environment Agency, 2010). This estimate, like $\beta$, tells us that for a significant backwater (and therefore storage) we need to have a small slope and larger depth. In other words, small $\beta$ indicates that the capacity to hold back a significant volume of water behind the dams is very limited. However, $\beta$ can also be large due to large h – which can lead to increased probability of failure, if $h > H$, so in Section 4 the model is applied to understand different configurations.

A small value of $\beta$ is also the first indication of why a large number of dams may be required to have even a noticeable effect on the discharge downstream. The small value here is an artefact of the assumed uniform width of the channel, but it is also consistent with the work of Metcalfe et al. [2017], where 57 leaky barriers were required in the 29km$^2$ Brompton catchment, along 4.7km area of the main stem and Ing Beck, before significant attenuation was achieved. It is likely that the locations for the dams may in reality be chosen to dam "reservoirs" that are wider than the average stream width, where the stream bed is particularly flat, or where there is capacity for significant overflow onto the floodplain, or additional off-line storage (e.g. Nicholson et al, 2019). We suggest that the contribution to the cross-sectional area due to the volume in the reservoir in (9) is therefore underestimated, and should be increased. Thus we modify the cross-sectional area to:

$$\tilde{A}(h) = wb + \lambda \frac{w(h-b)^2}{2Sl}, \quad H \leq h \tag{14}$$

where $\lambda \geq 1$ is this enhancement factor that accounts for a larger volume being stored behind the dam. In practice, this would have to be estimated for each dam location.

One potential concern with the above formulation is the discontinuity in the discharge-depth relation when the water depth reaches the bottom of the dam (Figure 4). This occurs in the model because the physics used to relate the depth to discharge is different in the two cases of free-stream flow (when we use Manning's law to describe turbulent drag) and flow under the dam (when we use an essentially inviscid formula for flow beneath a sluice gate). Mathematically, provided the discontinuity in flux involves a reduction as h increases, there should be no problem. As the water depth reaches the bottom of the dam, the flow past it suddenly reduces and the water quickly fills up behind the dam until the depth has increased to allow sufficient flow to balance the inflow from upstream. There can be problems, however, if the discharge suddenly *increases* when h increases past b which, from (8), occurs if the slope is sufficiently small, or n sufficiently large. If that occurs, we continue to use the frictional formula in the first case of (8) until the second formula gives a lower value for the flux.

### 2.6 Solution method

The system of equations (1), coupled with the expressions for $\tilde{Q}(h)$ and $\tilde{A}(h)$ in (8) and (9), is a system of non-linear equations for the temporal evolution of the water depths $h_i$. The equations are forced by the source terms $q_i$ and each equation is coupled the equations corresponding to the upstream edges of the network. The whole system is solved numerically using an ordinary differential equation solver in Matlab, using code that we have published in a repository as cited at the end of this paper.


## 3.0 Results

### 3.1 One-dimensional network

In this section we consider a simple example of the model, using the one-dimensional network shown in figure 2b. We suppose that each of the channel segments is the same (i.e. equal widths, lengths, and slopes), and the discharge in the final

segment is of most interest for the community requiring protection. For these calculations (and all others shown in this report) we use the original flux and area formulas (8) and (9), with the enhancement factor described in (12).

The model is forced with a `storm' input in the form of a hydrograph based on a simplified Gaussian functional form, as an approximation to a typical design storm estimated using unit hydrograph approach (used in the application to the real case in Section 4).

$$q(t) = q_0 + q_{max} e^{-t^2/\sigma^2} \qquad (15)$$

Where $q_0$ is a baseline inflow (groundwater flow into the channel, say), $q_{max}$ is the peak flood inflow at time $t = 0$, and the flood is spread over a time period $\sigma$. In figure 5 we compare the resulting modelled discharge in each channel segment

between the case of no dams, and the case of having a dam on each segment (we use only five segments for ease of illustration; using more segments allows for greater potential of reducing the peak discharge).

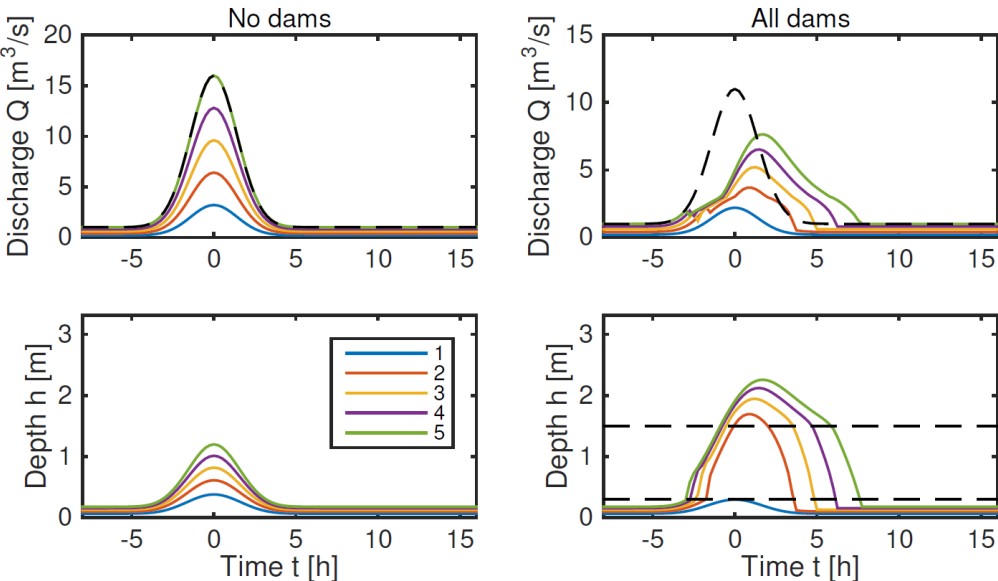

**Figure 5: Solutions for a one-dimensional 5-node network as in figure 2b, forced by uniform inflow to each node $q(t) = q_0 +$**
**$q_{max}\exp(-t^2/\sigma^2)$, with $q_0 = 022$ m³s⁻¹, $q_{max} =3$ m³s⁻¹, and $\sigma=2$ h. Parameter values are as given in section 2.3, together with $\lambda= 50$, $b =$**
**0.3 m and $H = 1.5$ m. Left-hand panels show the response with no dams, when the peak discharge is almost identical to the peak**
**cumulative inflow, shown by the dashed line in the upper panels. Right-hand panels show the response if a dam is included on each**
**of the 5 reaches. The dashed lines in the lower panel shows the heights of the bottom and top of the dams.**

Figure 6 shows an example when the input has a double peak. In this case, as might be expected, the dams are less effective at reducing the height of the second peak, because they are already holding back a lot of water and have less capacity to store

and delay water for the second storm. This indicates that testing of the performance of NFM, or any risk reduction measures, should potentially consider testing resilience against real storm series or double peaks and not simply single peaked storm events, as are commonly assumed in practice when considering flood storage design analysis.

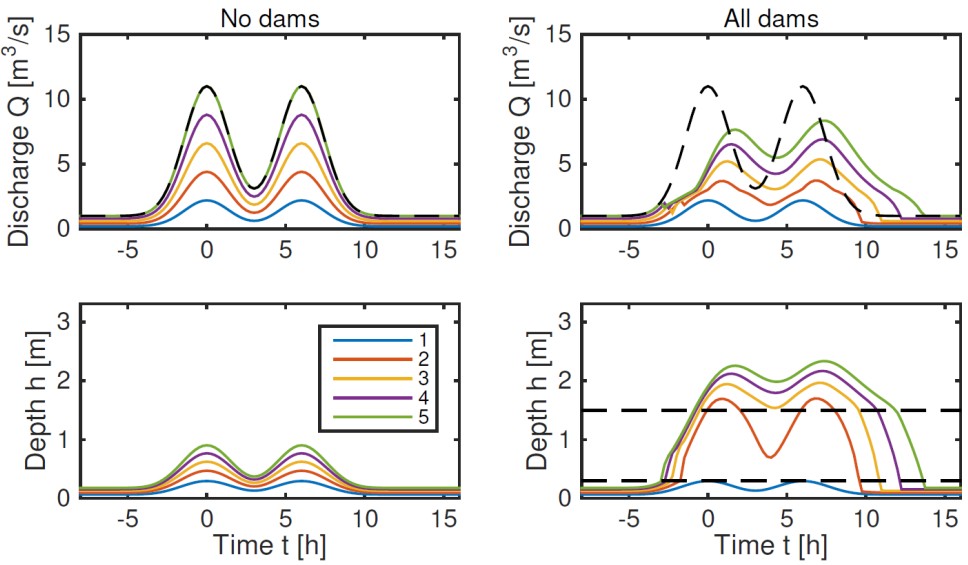

**Figure 6: Solutions for a one-dimensional network as in figure 2b, forced by a double peaked input to each node. Parameter values are as in figure 5.**

### 3.2 Two-dimensional network

Here we consider a simple two-dimensional network as shown in figure 2c, which reflects a more likely pattern given the dendritic nature of channel formation in headwater catchments. There are more interesting questions to consider about the positioning of dams in this case. For example, if one has funding to build a certain number of dams, which of the channel segments are the best ones on which to put them? Putting them on the central trunk is likely to ensure that they are used

(performance issue 1), but also means that they may more easily overspill and lose their effectiveness. They may also be more susceptible to cascade failure (performance issue 3 - discussed in the next section).

In figure 7 we show two examples of the response to a flood input of the form given by equation (15). In the first case, 4 dams are placed on the main trunk (nodes 1-4), whereas in the second case 4 dams are placed on the upper branches (nodes 5,6,9,10). The discharge from the final segment (node 4) is plotted, along with its maximum value. Both dam placements

have the effect of slightly delaying and reducing the peak discharge, with the second design being marginally more effective. This is because the dams near the bottom of the central trunk are over spilling and losing their effectiveness, whereas the dams on the side branches are all having a significant effect.

However, for different sized floods or realistic spatial patterns of extreme rainfall (see Hankin et al., 2017a), the optimal arrangement can vary. Unfortunately, there does not appear to be a clear rule for the most effective dam placement, even in

this simple example, where the resilience of distributed NFM in terms of temporary storage and tree-planting were tested against different storm extremes having spatially realistic patterns (Lamb et al., 2010). In this network study, the 'on average' performance of one particular system of NFM was tested allowing for utilisation and the risk-reducing or risk-increasing impacts of changes to tributary synchronisation (performance issue 2), using average annual losses as the integrated measure of risk reduction. However, the high-resolution model, with 180 million cells, took over 26 hours to run

so only 30 extreme events were simulated with and without NFM measures, and *alternative* spatial strategies were *not* tested to understand which were more advantageous. Simplified network analyses such as those presented here could be used to rapidly explore such spatial strategies, without resorting to highly complex, and relatively slow models.

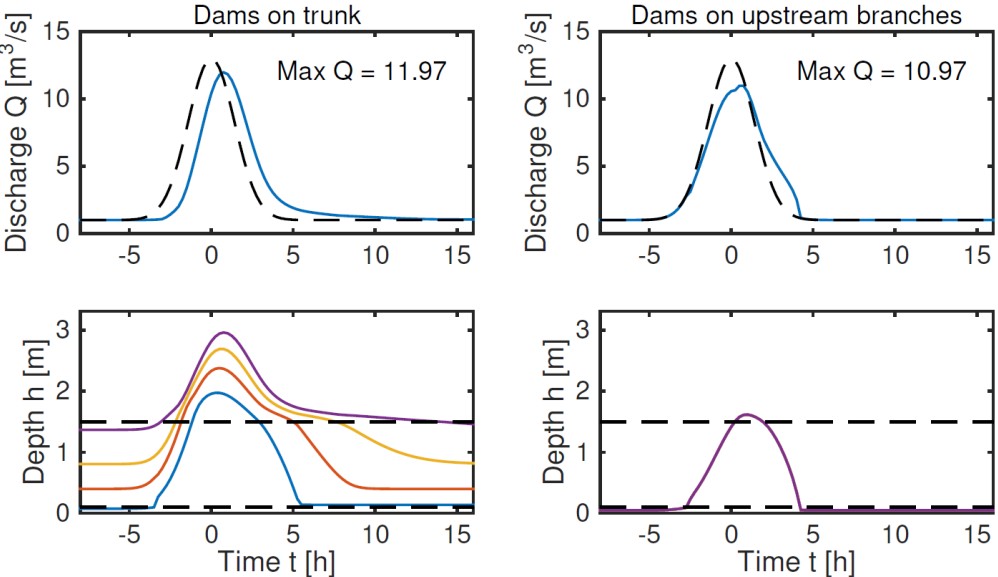

**Figure 7: Solutions for a one-dimensional 5-node network as in figure 2b, forced by uniform inflow to each node q(t) = q0 + qmaxexp(-t²/σ²), with q0 = 0.125 m³s⁻¹, qmax =3 m³s⁻¹, and σ=2 h. The upper panels show the discharge downstream (node 4), and the lower panels show the water depth at the 4 dams (nodes 1-4) for the case on the left, and the identically-behaving nodes 5,6,9,10 for the case on the right). Parameter values are as given in section 2.3, together with λ = 20, b = 0.1 m and H = 1.5 m.**

### 3.3 Failure mechanisms

One of the potential risks of installing many dams in a catchment is the possibility that they all collapse in sequence, creating a flood surge that is larger than would have occurred if no dams had been installed at all. Provided each dam stores only a small reservoir of water, the collapse of one dam on its own should not be catastrophic. But if the collapse of one dam causes others further downstream to collapse too there is the obvious danger of the surge escalating. This risk may be an important factor in deciding the best placement of dams (perhaps outweighing the efficiency of peak-flow reduction under `normal' operating conditions).

The main method suggested for analysing this risk is to run an ensemble of simulations of flood events, assigning a failure depth to each dam using the probability distribution suggested by a fragility curve. Ideally, this ensemble should include a range of storm conditions too. Such analysis could in principle use dynamical weather models or rainfall records to construct statistical models and then sample from the modelled joint (spatial) distributions of rainfall forcing. Both approaches have been considered in the context of reviewing flood resilience in the UK (HM Government, 2016) and for flood risk analysis over large and complex infrastructure networks (Lamb et al., 2019) This type of spatially-structured risk analysis can be expensive, and so it may be desirable to establish some rules of thumb about which dam placements are more, or less, at risk from cascade failure.

Knowledge gained from this type of analysis might be used to plan for the size and strength of dams that should be built at different locations. For example, it could be that certain locations are particularly prone to collapse (downstream of merging tributaries for example), and building one stronger `buffering' dam could significantly reduce the risk of a cascade.



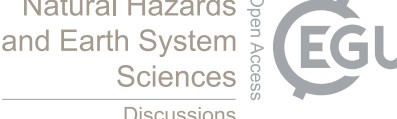

As an example of cascade failure in the network model, we return to the one-dimensional example shown in figure 2b. We impose a regular storm inflow to the upstream node of the form given in (15) and examine an ensemble of 50 possible system states (describing different combinations of survival or failure of the individual dams). Each of the dams is assigned a critical water depth $h_{ci}$ such that when $h_i > h_{ci}$ the dam collapses; the critical depth is sampled from a normal distribution

with mean 3.5m and standard deviation 0.5m (the top of the dam is at 1.5m so dam collapse usually occurs when the dam is already submerged). The results are shown in figures 8 and 9. Figure 8 shows the peak discharge $Q_{max}$ at the downstream node for each simulation.

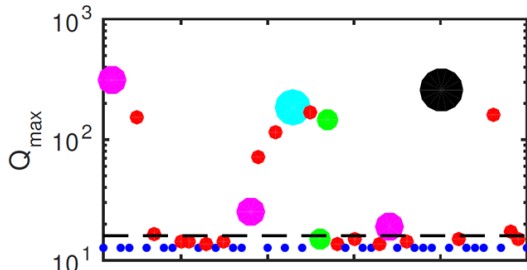

**Figure 8 Maximum discharge at the downstream node for an ensemble of runs (indexed along abscissa) on the one-dimensional**
**network in figure 2c, forced by the same upstream inflow q(t) = q0 + q$_{max}$exp(-t$^2$/σ$^2$) , with q0 = 1 m$^3$s$^{-1}$, q$_{max}$ = 15 m$^3$s$^{-1}$, and σ = 2 h. The size and colour of the dots indicates the number of dams that failed during each realisation, and the dashed line shows the peak discharge in the case that no dams are installed. Parameter values are as given in section 2.3 except with l = 1000 m, S = 0.005, and λ = 20, together with b = 0.3 m and H = 1.5 m.**

Figure 8 is coloured by the number of dams that are predicted to collapse within each ensemble member; small blue dots indicate that no dams failed, and since the forcing is identical in each case, the peak discharge in this case is always the same. It is lower than what the peak would have been in the absence of any dams, so the dams are proving effective in these cases. Larger dots correspond to more dams having failed. In most of the ensemble members only one dam collapses, and the peak discharge recorded downstream is strongly dependent on which one fails (the red dots in figure 8). A larger peak

occurs if the collapsed dam is further downstream, since if an upstream dam fails (and importantly does *not* precipitate a cascade of downstream failure) the sudden release of water from that dam is buffered by the dams further downstream. If, on the other hand, a single dam failure leads to further collapse of two or more dams, the peak discharge can be much larger. In one example, all five dams collapse in quick succession, and the time series of this example (the black dot in figure 8) is shown in figure 9, where it is compared to an example with no failure. We have found that the pattern of failure

in this one-dimensional model, including the likelihood for cascading failure, depends heavily on the assumed dam sizes, critical water depth distribution, and the magnitude of rainfall events.




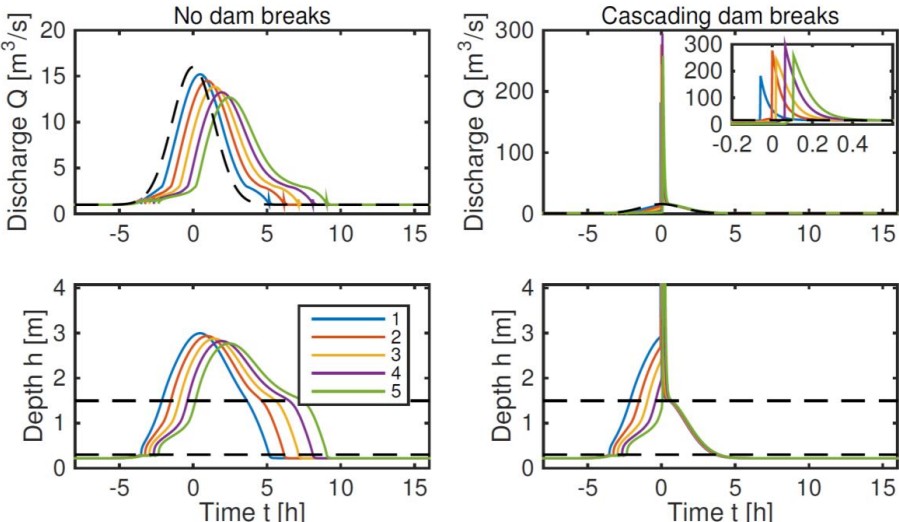

**Figure 9: Example from an ensemble of runs on the one-dimensional network in Figure 2b, forced by the same upstream inflow $q(t) = q0 + q_{max}exp(-t^2/\sigma^2)$, with $q0 = 1$ m$^3$s$^{-1}$, $q_{max} = 15$ m$^3$s$^{-1}$, and $\sigma = 2$ h. The dams have critical failure water depths $h_{ci}$ drawn from a normal distribution with mean 3.5 m and standard deviation 0.5 m, and shown by the coloured dashed lines in the lower**
**panels. In the case on the left, no dams fail, whereas in the case on the right (when the uppermost dam is particularly weak), they all fail in a cascade. Parameter values are as in figure 7.**

As a second more instructive example of cascade failure, we revisit the herring-bone network in figure 2c. We consider the two possible placements of 4 dams that were discussed earlier; either along the main trunk (nodes 1-4) or on the upstream side branches (nodes 5,6,9,10). In figure 7 we found that there was relatively little difference in the peak discharge measured

at the downstream node with these different placements. However, in figure 10 we see that the first case is much more at risk from cascade failure. This figure shows the peak downstream discharge in an ensemble of simulations, with the failure depths for each dam being different each time. The failure depths are sampled from the same distribution in each case.

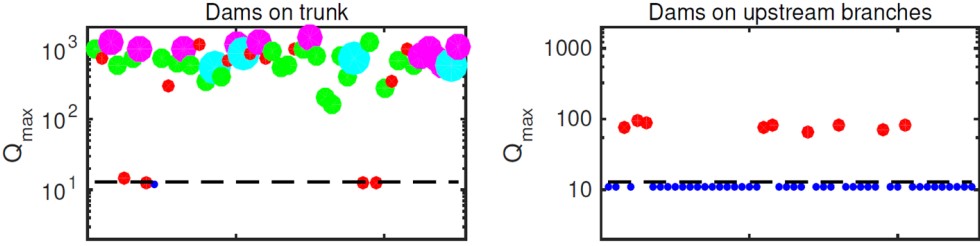

**Figure 10: Maximum discharge at downstream node for an ensemble of runs (indexed on abscissa) on the two-dimensional**
**network in figure 9, forced by uniform inflow to each of the 8 branch nodes $q(t) = q0 + q_{max}exp(-t^2/\sigma^2)$, with $q0 = 0.125$ m$^3$s$^{-1}$, $q_{max} = 1.5$ m$^3$s$^{-1}$, and $\sigma = 2$ h. Each dam is assigned a failure depth $h_{ci}$ drawn from a normal distribution with mean 3.5 m and standard deviation 0.5 m. In the first case, dams are placed on the four trunk segments (nodes 1-4), while in the second case they are placed on the upstream side branches (nodes 5,6,9,10). The size and colour of the dots indicates the number of dams that failed in each realisation, and the dashed line shows the peak discharge in the case that no dams are installed. Parameter values are as in figure**
**7.**

In almost every run with dams on the trunk, we see cascade failure occurring so that 3 or 4 of the dams collapse; this leads to extremely high (though short-lived) peak discharge. In contrast, when the dams are placed on the side branches, there is no possibility of cascade failure (the individual branches do not communicate with each other) and it is unlikely that more than one dam collapses. Thus, although each of these dam placements is similarly effective at reducing the peak discharge, there



may be strong reason for preferring the second design that places them on the upstream tributaries because this configuration is a more resilient system (even though the resilience of the individual dams is the same in both designs). The large surges predicted in the simple network model when multiple dams fail have some support in the literature. For example, Hillman [1998] describes a June 1994 outburst flood in central Alberta, Canada, releasing 7,500m$^2$ of water and a flood wave 3.5

times the maximum discharge recorded for that creek over 23 years. Although not reported as a cascade failure, large trees and debris from older beaver dams were carried further downstream, and five hydrometric stations downstream were destroyed.

**4.0 Application of model to Penny Gill, West Cumbria**

We consider the application of the model to a site on Penny Gill, West Cumbria (figure 1). The geometry of the network of

leaky dams installed by the West Cumbria Rivers Trust is shown on the left of figure 11, with the inflow hydrograph on the right. The inflow hydrograph has this time been based on a 100 year return period design hydrograph using Revitalised Flood Hydrograph or ReFH version 1 (Kjeldson, 2007), which is based on a unit hydrograph approach assuming empirical relationships with local catchment descriptors such as slope, annual average rainfall from the Flood Estimation Handbook (IH, 1999). ReFH also includes a losses model that accounts for hydrology of soil types and gives the hydrograph in this

instance a slight tail due to slower baseflow contribution. It should be noted that the peak flow for the 100 design year event is relatively small but is likely to be under-estimated owing to contributions from old coal measures that are known to generate additional flows during times of prolonged rainfall.

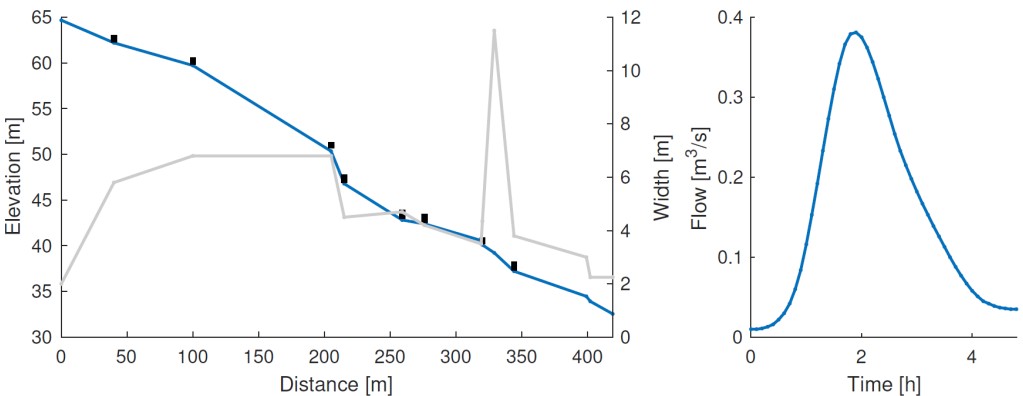

**Figure 11: Left: Geometry of the Penny Gill network with given position of dams and widths labelled. Right inflow hydrograph**
**based on ReFH approach described in the text.**

The model is as described in section 2, with mass conservation for each network given by equation (1) as before, although in this example water is all fed into the upper most segment of the network according to the input hydrograph $q_{in}(t)$ shown on the right of figure 11. Discharge and cross-sectional area are related to water depth at the dams $h_i$ by the functions $Q_i =$

$\tilde{Q}(h_i)$ and $A_i = \tilde{A}(h_i)$ given in by dimensionless equations (11-13) in section 2. The only changes are that λ=1, and variable lengths, widths and slopes are being used for different segments, allowance for which was already made. The very wide segment in figure 11 reflects an area of channel well connected with a relatively wide depression.

A model calculation using the measured values for the dam parameters (bottom height b and top height H) showed very little delay or reduction in the peak of the downstream discharge hydrograph relative to the input hydrograph. This is because the


dams were hardly coming into operation at all. However, at a recent site-visit there was siltation and noticeably small slot heights for many of the leaky barriers, over much of the width of each construction. Therefore, an example was simulated with the bottom heights lowered to 0.02m, as shown in figure 12, which ensures more of the storage comes into use. It should be noted that for this 1-d model, the slot heights represent an *average* across the width of the barriers, which

5 represents a further approximation.

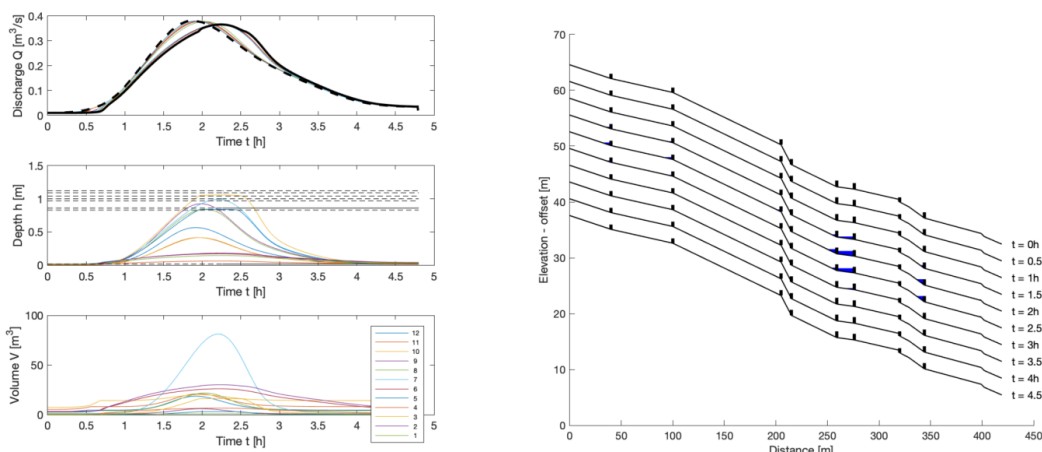

**Figure 12: Model run using the given measured dam locations and bottom height of 0.02m. Note that each segment has a different width, which is not shown on the diagram, but which can lead to different rates of filling of the region behind different dams. We take a Manning roughness of 0.1 s m$^{-1/3}$. The reduction peak discharge is 96% of the inflow, and the maximum volume stored is**

10 **245 m$^3$.**

We can use the network model to quickly explore alternative arrangements of the dams, and examine whether there are general rules about how to site the dams that could lead to more efficiency. To do this we break the stream into 20 segments, and allow for the possible siting of a dam on each one. All such dams are assumed identical, with bottom height b = 0.02 m

15 and top height H = 1 m. An example calculation with dams on every section is shown in figure 13. The issue of under-utilisation is clear from this example. Note also that although more water is stored than in figure 12, the reduction in peak discharge is actually slightly less. This is because the dams are already full by the time that the peak now happens, and illustrates the subtleties involved in deciding how and where to place the barriers.




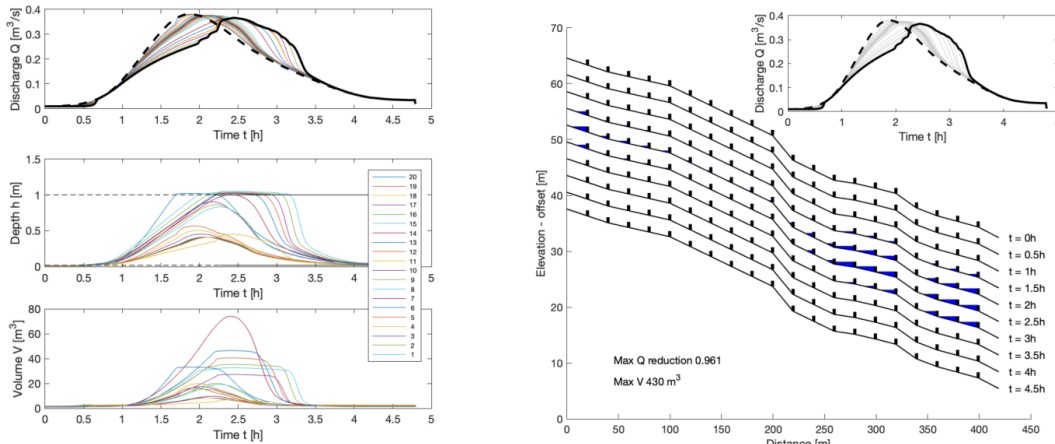

**Figure 13: Model run with more leaky dams. The reduction peak discharge is 96% of the inflow, and the maximum volume stored is 430 m³.**

5    We also consider randomly siting 8 dams on the 20 segments, to analyse which positions work well. Examples of some of these are shown in Figure 14. Some are considerably better at reducing the peak flow than others (though none are particularly good - simply because there is not enough storage overall to reduce the peak discharge substantially).


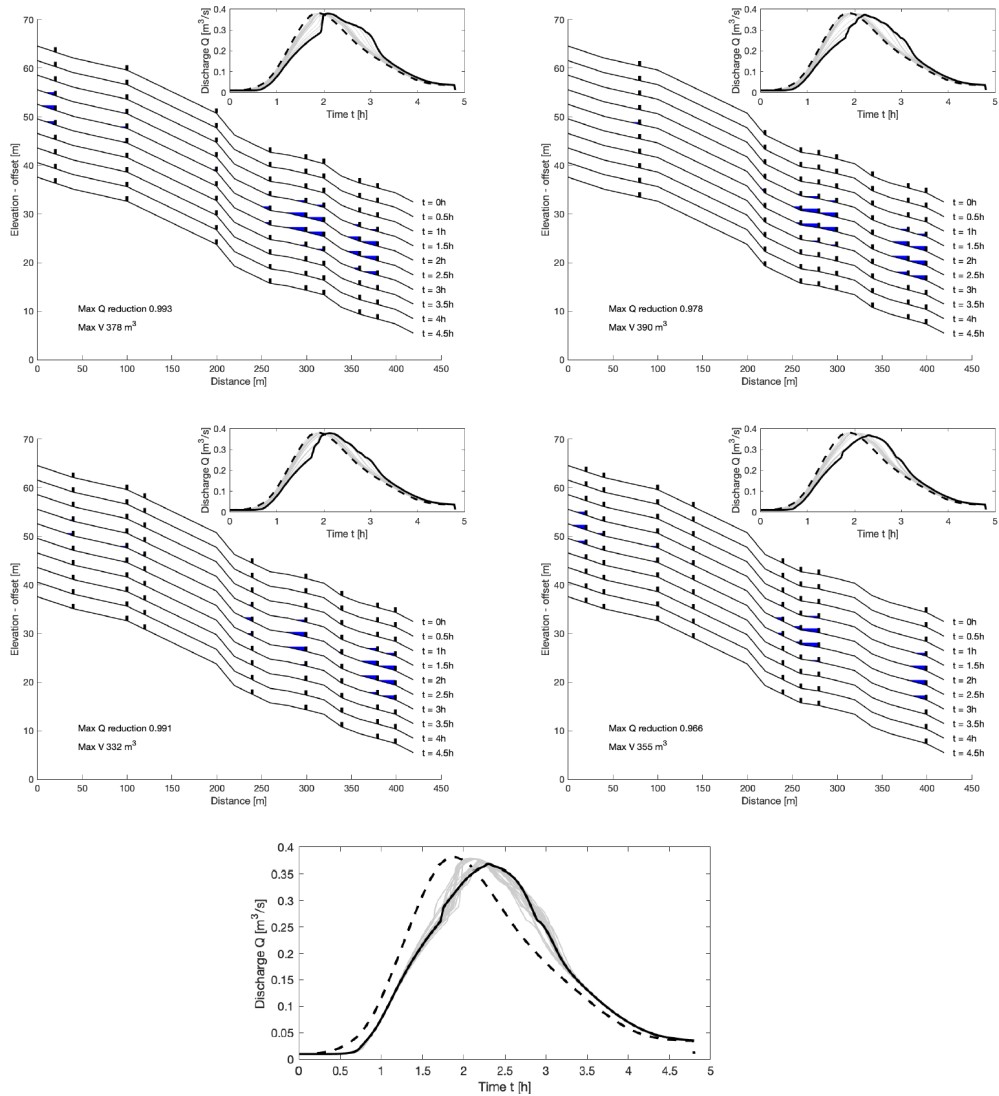

**Figure 14: Example model runs with random arrangements of 8 dams. The lower panel shows the downstream discharge for each of 20 different arrangements, compared with the input hydrograph (dashed), and with the one that gives the greatest peak reduction (97%) highlighted in black, which is the lower right of the examples above and results in a maximum volume stored of 355m³.**

By choosing to site dams on shallower segments of the network, storage is enhanced and under-utilisation is avoided; an example is shown in Figure 15. This has almost as good performance as building dams on all 20 segments (compare with Figure 13). Note however that this is partly due to assuming identical dams. Since the water depth would typically be lower on steeper stretches, it would be natural to have a smaller slot under the dam there so that the dam becomes active. Nevertheless, the inability to store significant water behind a dam on steeper slopes means that such locations should generally be avoided.





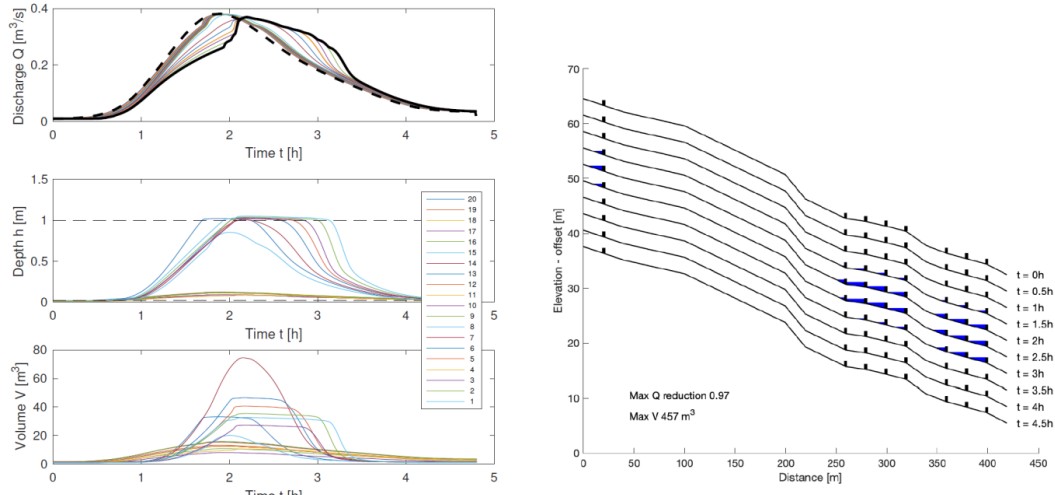

**Figure 15: Model run with 8 dams, chosen to be sited on segments with lower slopes. The reduction peak discharge is 97% of the inflow, and the maximum volume stored is 457 m³.**

What insights can we learn from this? A good general principle is to site the barriers in low-lying areas or regions where the upstream area widens, so as to provide more storage per barrier. The lower height of the barrier should be made sufficiently high that it does not start to dam water too early; if the dam has filled before peak inflow conditions, it serves no further useful purpose in reducing the flow downstream. Dams in locations with a large backwater region (which is characterised by the local value of β, although it is more easily interpreted just in terms of being shallow-sloping and wide) are worth building

higher, and making correspondingly stronger so as to avoid the risk of failure.

Such advice could be used at a much larger scale to supplement the relatively scarce advice for where to locate leaky barriers, which also tends not to include details of geometries, leakiness or slot heights. For example, in England and Wales, countryside stewardship grants can be applied for to support construction of leaky barriers and the website[3] advises that

leaky barriers should be sited on channels between 3 and 5 m, yet says nothing about slope, height and slot dimensions, which could be more of a determining factor in the effectiveness of potential storage. The dimensions of the slot height are also not clear, and vary in grey literature between 0.1- 0.3m, but in practice, are less (on average across a cross section) than this in locations like Penny Gill. A compounding factor is that there are many forms of leaky barrier or large woody debris dams (Addy et al., 2019) including placing large woody debris in channel, to the horse-jump type barriers in use in Penny

Gill, combined with engineered log jams which will also reduce passage of debris should a structure fail.

Probably more importantly, the final example demonstrates that this type of model can be effectively used to quickly try out different arrangements of dams and to assess which are likely to work best. Only one particular input hydrograph has been used here, and a more thorough analysis ought to consider different amplitudes and shapes and multiple peaks, since there

are likely to influence the effectiveness of the whole scheme. It would also be useful to test further failure scenarios, although the simplified 1d representation does not include log-jams that were also placed between some of the dams, which would help mitigate the risk of cascade failure explored in Section 3.

---

[3] https://www.gov.uk/countryside-stewardship-grants/rp33-large-leaky-woody-dams



### 5.0 Conclusions

We have formulated a network model for a catchment area that allows for simple exploration of the effectiveness of different dam placements and designs and is sufficiently cheap to solve that it may be useful in analysing risks that require a large ensemble of simulations. We have applied the model for relatively small idealised and real systems of around 10-20 dams,
but its computational simplicity means that it would readily scale to consider much larger systems at little additional cost. Based on the analyses presented, we can make four practical conclusions focussing on the three performance issues highlighted, those of utilisation, synchronisation and failure or cascade failure:

- A large number of dams are needed to have any significant effect on the peak discharge downstream based on scale analysis alone, especially in reaches with steep gradient. When estimating storage requirements, it is not sufficient
to simply estimate the total storage capacity in relation to the volume under the hydrograph for a set of NFM measures that are distributed around the network. Network analysis is required that also permits the assessment of the integrated impact of: **dynamic utilisation of storage**; **drain down between events** (tested by simulating on multi-peaked events); and **changes to flood-peak synchronicity** on overall risk reduction. This has been measured in terms of reduction to peak flow hydrograph at the bottom of the network between the pre- and post-NFM
situation, providing an integrated measure of the effectiveness of the system of NFM features.

- The dams should be located in places with the potential to store a reasonable volume of water (in wide reaches of the channel), although with consideration that the loading on each structure is not excessive. With reference to a real-world example at Penny Gill, we have used the network model to highlight how locating dams in areas with wider channel width and low slope is more effective and are worth building higher, and making correspondingly
stronger so as to avoid the risk of failure.

- These conclusions on placement help, in part, to understand whether there are any benefits from making an effort to place dams strategically, seeking an optimal network configuration, or whether it may be justifiable to install them opportunistically, or even randomly. The analysis indicates that, for the relatively simple system at Penny Gill, when considering potential dam sites at up to 20 locations, approximately 50% of effort could be saved in
construction, costs and later maintenance, if fewer dams are placed more selectively. It remains to be seen whether there are any broader advantages at large scales (>100km$^2$, say) of macro-scale strategies, such as targeting one whole side of a valley or another. The approach demonstrated here enables such analysis to be carried out.

- Cascade failure is a risk when dams are placed along a main artery and the risk may be lessened by spreading dams around tributaries. There are very large uncertainties in the fragility assumptions leading to failure, although here
water depths of an order of *twice* the barrier height were used, which would present a considerable loading. Despite the uncertainties in the probability component of risk, the potential consequences, which appear to be evident in historical events in natural systems, should highlight the need for robust barrier design supported by good engineering design of leaky-barriers. For the case-study of Penny Gill investigated here, the placing of large woody material within the channel between dams is another good risk reduction strategy.

We envisage future risk assessments using this network approach at larger scales, taking into account additional factors including: uncertainties in geometry, roughness parameterisation, spacing, fragility assumptions, a wide range of spatial configurations of NFM measures, and a wider range of feasible storm types, durations and probabilities. These are all required not just for NFM, but also for improved integrated flood risk management, if we are to answer the types of simple
questions that communities need to answer, such as: "With a limited budget, what's the best approach for integrated flood risk management?", or, "does spatial configuration even matter at a larger scale?". We hope our conclusions here start to address such questions, but future analyses would also be better constrained with a more detailed understanding of the



fragility of different types of barriers. More formal fragility curves can be directly generated (Lamb et al., 2019) based on analysis of observations of survival and failure of dams, if such data is recorded for the growing number of leaky barriers.

## 6 Author Contribution

Barry Hankin wrote the paper with major contributions from Ian Hewitt and Rob Lamb. All authors helped in developing the
model at the Maths Foresees Study Group.

## 7 Acknowledgements

This work has been supported by NERC Grant **NE/R004722/1**, the EPSRC Maths Foresees network and the JBA Trust project W17-6962. The model was initially developed at the EPSRC-funded 'Maths Foresees' Study Group in Cambridge, April 2017, where this particular challenge was sponsored by the JBA Trust. The problem was worked on by Barry Hankin,
Ian Hewitt, Graham Sander, Sheen Cabaneros, Federico Danieli, Giuseppe Formetta, Raquel Gonzalez, Michael Grinfeld, Teague Johnstone, Alissa Kamilova, Attila Kovacs, Ann Kretzschmar, Kris Kiradjiev, Sam Pegler, Clint Wong.
We are grateful to Onno Bokhove for introducing the problem to the study group. We are also grateful to the West Cumbria Rivers Trust for access to Penny Gill, along with MSc student Luke Stockton for assistance surveying the leaky barriers.

## 15 Code Availability

The Matlab scripts developed are available on the JBA Trust GitLab repository:
https://gitlab.com/jba-trust/leaky-barrier-network-analysis

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
