# Peer review of "A risk-based, network analysis of distributed in-stream leaky barriers for flood risk management"

_Natural Hazards and Earth System Sciences, 2019_

## Referee Comment (RC1) · Anonymous Referee #1 · 23 Mar 2020

**Review**

**A risk-based, network analysis of distributed in-stream leaky barriers for flood risk management**

Barry Hankin, Ian Hewitt, Graham Sander, Federico Danieli, Giuseppe Formetta, Alissa Kamilova, Ann Kretzschmar, Kris Kiradjiev, Clint Wong, Sam Pegler, and Rob Lamb

The text has been analyzed twice:

1. Iteration: questions and error detection
2. Iteration: cross checking fulfilling the declared aims and answering the scientific questions

**Iteration 1:**

<questions and error detection ready, but not typed in, delivered if necessary>

**Iteration 2:**

Deliveries and scientific questions as defined by the author Barry Hankin.

**Check of deliveries**:

- Delivery of network based model. Fulfilled.  (215 lines matlab- code; 25 lines calculation, 190 lines I/O and plotting)
- Rapid assessment. Fulfilled.
- Design advice. Not fulfilled. (No quantification, the given design proposal is not derived from the network model)
- Understanding of effective risk reduction strategies. Partially fulfilled. (advices are given, but on a predictable level)

**Check of scientific questions**:

- Effective dynamic utilization of storage at network dam placements. Partially fulfilled. (There is no criteria given how effectiveness is defined in this context. The effectiveness is not exactly quantified. The effect  for the given examples is low. The effectiveness is not linked to the given design flood.)
- Identification of placements that reduce the risk of (cascade) failure. Partially fulfilled. (just qualitative analysis possible, because both dam leakage and fragility are  not known.)
- Do small-scale interventions using NFM (Natural Flood Management) combine to create a large scale benefit at large scale?  Not fulfilled. (No definition what a large benefit in a large scale is. No model investigation carried out analyzing this particular question. The models used are by far too small.)

- Reliability and performance of NFM-measures under plausible hazards. Not fulfilled. (No plausible hazard given. The test case with Qmax= ca. 10m3/s is completely synthetic. The case study area is not given, but the small N100 resulting in Qmax = 0,5m3/s indicate a very small catchment area, which probably is not representative.)
- Resilience of a network of NFM-measures. Not fulfilled. (Resilience is neither defined nor addressed in the results.)

Missing:

- Definitions on effectiveness, resilience and "large benefit at large scales"
- Hydrological aspects (f.i. definition of design hydrograph, impact of convective or orographic rainfall events on efficiency)
- Ecological aspects
- Arguments for the development of the network model (there are well-proven 1d-tools as MIKE11 or HECRAS)
- Validation of proposed network model for normal flow conditions and dam-break

Facit: Very basic; right questions but weak method.

---

## Referee Comment (RC2) · Paul Quinn (Referee) · 27 May 2020

Review of …

**General Comments**

I really enjoyed this paper. It was novel, innovative and thought provoking.

I do have a number of fundamental detailed points that I will outline below.

Overall I am concerned that the RAF/Flood features being shown are not really good features to build or simulate. However, the theoretical analysis suggested that small networks of barriers can have a huge impact on flood reduction. This needs a number of clarification points. Firstly the feature shown are small and create small volumes of temporary storage. The features are trapped within the channel and hence seem to have little capacity to store flow. Hence the conclusion must be that it is the roughness that is slowing the flow???? This type of feature works better when the water is forced onto a floodplain and into extra storage areas. So I would be suggesting that RAF design is key, i.e. using the barriers in combination with other storage and velocity reducing zones, e.g. shallow flow across floodplains. A criticism of the Metcalf work is that the features did not allow flow onto to the farmland next to the channels.

However, let's go with the network of within channel features and discuss that.

One more 'picky' thing the author uses the term 'we' a lot. In the past this would not be allowed but I know we live in enlightened times and we can now use 'we'. I do think we have an overzealous use of the term 'we', especially when occurs 4 or 5 times in paragraph.

'We have formulated a network method…' could be 'A network method has been formulated…'

I think earlier work on network models could be included and referenced. E.g Nicholson et al., 2019, Quinn et al., 2015 which was included in the WWNP report. I think Bhoko et al., 219/20? Could be added.

The work of Nicholson included observations and this study has used a theoretical approach and this may need to be highlighted.

**Detailed Comments**

Bhoko comments on beavers and the impact on flood flow. It is also important to stress that beaver dams are full of water all the time. So that is why they are useless to flooding!

I think your model is not simulating a typical leaky dam, even though figure 1 is a little leaky.Most leaky dams are usually large woody debris that are very porous/leaky. You are thus assuming that all leaky dams operate as a sluice gate. This may not be a good assumption. But, let's go with the impact of a network of sluice gates on flood flow.

Fig 2 is crucial as is section 2.2. What is the size of the catchment? Add a scale to fig 2. I know you want it to be a dimensionless model, but surely a catchments size would help. You need to state the density of barriers. I deduced later that the model had 100m lengths between nodes.

Later you show the model resulting as being based on Fig 2b, so this means the length of channel is 500m, This means the catchment is about 1km2? SO, how do you get a flow of 15 m3/s (fig 7). This

would be a flow rate for a 10-20km2 catchment. That catchment area would require 100 RAFs. Can the authors sort this out or address my misunderstanding.

The model result then suggest a huge reduction in flow caused by 5 barriers?? 15m3/s down to 6m3/s. But the storage behind these barriers must be tiny. Surely they would be overwhelmed by such high flows. SO I may have missed something. SO what is causing the reduction? Is it roughness? Could the authors show some sensitivity analysis of Mannings 'n' on the model output?

I like the analysis introducing branches, but again I do worry about scale and flows. The point being made about branches have less danger is good, as this reflects that a scale appropriate positioning of RAFs is needed, as they work better and have less problems in smaller channels.

Overall I was less impressed with the analyses and cascade failures. I am not addressing the analysis which is good, but a NFM team would either design in the failure so that lower dams trap debris or if there is a threat than a debris traps would be built.

SO, now we look at Penny Gill, the flow rate is 0.4m3/s. This analysis section is interesting. However, it reinforces my point about the flow rates used above. You really have a chalk and cheese comparison. The features in Penny Gill should have been better designed, i.e. to use zones with active floodplains (if appropriate) or maybe the wood should have been distributed over larger areas to create more roughness, thus could be more tree planting in the channel, e.g. willow.

I hope the authors do not think I am being over critical. I really did enjoy the paper. I may not fully understand the model, the scale, the flows, the volumes and the RAF density assumptions. If you sort this out then this paper can be published. If it is a simple clarification of the model and the assumptions then this would be just a minor edit. If there is a fundamental issue with the flow and the impact then major corrections are needed.

---

## Author Comment (AC1) · 6 Jul 2020

**Author Comments to public discussion of "Network performance of leaky barriers NHESS 2019-394 Discussion Paper".** B. Hankin, R. Lamb, I. Hewitt, 01-07-2020.

Dear Editor and Reviewers,

Thank you very much for the review comments on our NHESS discussion paper on a network performance model for leaky barriers.  We have provided a response to each reviewer comment in the table below and think the proposed changes will lead to an improved and clearer manuscript.

This paper came out of an applied mathematics group challenge to understand the performance of a network of nature-based, leaky-barriers and to establish a fast model capable for testing many configurations at larger scales as a proof-of-concept.  Our preliminary investigation uses a synthetic case with *extreme* flows, specifically chosen to stress-test the network and produce failures (and cascade failures) of the leaky barriers. These test cases also *incorporate substantial storage* (which may not be apparent from a first look at the equations), made possible using a convenient factor, λ, which may have been overlooked in the review.  We are very much in agreement regarding the need for leaky-barrier type solutions to connect to as much storage as possible and the formulation provided with a solution in one-dimensional format permits such testing for a wide range of situations.

We first use the synthetic case to establish that the model can account for open channel flow with variable storage in 2d networks of leaky barriers whilst incorporating under-flow, over-flow, porous through-flow, barrier failure and cascade failure. We demonstrate its flexibility in application to a real network of existing barriers, and consider that the paper and model we have provided on *gitlab* will help others to answer pressing questions on configuring spatial strategies for large networks of leaky barriers.

**We first make some general comments in response to Reviewer #1 and then tabulate our response to each reviewer point in the table further below**

**Reviewer #1**

We have assessed the comments made by Reviewer #1. We have looked to address the constructive suggestions for improvements in the reviewer's comments; our proposed responses are given in a table that follows. We also have some general comments in response to this review, where we denote text taken from the reviewer's commentary in red.

The reviewer structured their comments around a cross-check of "Deliveries and scientific questions as defined by the author Barry Hankin."

We believe that in formulating their "check list" the reviewer has gone beyond the actual objectives we stated, as we explain below. Whilst we would concur that the objectives suggested by the reviewer are all of interest, they do not altogether represent our objectives in this paper.

**REVIEWER #1 Check of deliveries:**

- Delivery of network based model. Fulfilled. (215 lines matlab- code; 25 lines calculation, 190 lines I/O and plotting)

Agreed.

- Rapid assessment. Fulfilled.

Agreed.

- Design advice. Not fulfilled. (No quantification, the given design proposal is not derived from the network model)

We did not set out to offer design advice, believing that this would be premature. We are not aware of having set out any specific "given design proposal". Rather, our hope is that the paper demonstrates a method that could be applied further to inform design advice with respect to the configuration of systems of leaky barriers (as opposed to construction advice about individual assets).

We think the wording of our original text in places does not reflect this position clearly enough and propose to amend it accordingly. The first sentence in the Conclusions sums up our position "We have formulated a network model for a catchment area that **allows for** simple exploration of the effectiveness of different dam placements and designs and is **sufficiently cheap to solve that it may be useful** in analysing risks that require a large ensemble of simulations." (with emphasis added here to stress that we see this as enabling the application to inform design advice, rather than delivering design advice *per se*).

- Understanding of effective risk reduction strategies. Partially fulfilled. (advices are given, but on a predictable level)

This was not one of our stated objectives as written in general terms by the reviewer. Although it can be inferred that the analysis of NFM systems does, overall, seek to understand effective risk reduction strategies, our objectives stated on page 3 were in fact more specific, namely:

1) We want to understand NFM features as systems of assets, and to assess those systems within a risk-based analysis that considers the whole-system performance in terms of risk reduction
2) We wish to understand the impact of different spatial configurations of the leaky barriers, taking into consideration three possible performance issues:
    a. under-utilisation / redundancy
    b. undesired synchronisation
    c. structure failures including cascades

**Check of scientific questions:**

- Effective dynamic utilization of storage at network dam placements. Partially fulfilled. (There is no criteria given how effectiveness is defined in this context. The effectiveness is not exactly quantified. The effect for the given examples is low. The effectiveness is not linked to the given design flood.)

This is one of stated objectives (2a above). We agree it has been fulfilled in part and propose amendments to clarify how we define effectiveness (please see table).

- Identification of placements that reduce the risk of (cascade) failure. Partially fulfilled. (just qualitative analysis possible, because both dam leakage and fragility are not known.)

This is not one of our stated objectives. Our objective was to understand the impact of different spatial configurations of barrier networks, including the possibility of single and cascade failures (2c, above).

This could of course help to identify placements that reduce the risk, as we have discussed; however, we did not set out to identify specific placements (in an optimisation sense) as one of our objectives, only broad strategies.

We disagree with the reviewer's assertion that the analysis is qualitative. The results are very clearly quantified in the graphs plotted in Figures 8 and 10. Dam leakage and fragility are explicitly parameterised (i.e. quantitatively) in our model. The selection of suitable values for the leakage and fragility functions is of course a different question; in the discussion and conclusions we comment on the desirability of calibration in future applications of the model.

- Do small-scale interventions using NFM (Natural Flood Management) combine to create a large scale benefit at large scale? Not fulfilled. (No definition what a large benefit in a large scale is. No model investigation carried out analysing this particular question. The models used are by far too small.)

This was not one of our stated objectives.

It is an important, broader question that we raised for context in the introduction to the paper. We also comment in our discussion and conclusion that our model could be helpful in exploring this question further. But we did not set out to tackle scaling in this paper. Rather, we see the paper as a precursor step to demonstrate a model that is capable of scaling, which could be applied further as we mention in the discussion and conclusions.

- Reliability and performance of NFM-measures under plausible hazards. Not fulfilled. (No plausible hazard given. The test case with Qmax= ca. 10m3/s is completely synthetic. The case study area is not given, but the small N100 resulting in Qmax = 0,5m3/s indicate a very small catchment area, which probably is not representative.)

We agree that the plausibility of the test cases require further discussion and clarification. We have given details of proposed amendments in our response to Reviewer #2, who raised some specific questions about this issue. Reviewer #2 identified some potential for misunderstanding about how additional channel/floodplain storage is represented in the equations, which we propose to clarify through amendments (see table).

We cannot agree with the comment that the small catchment area in Penny Gill is "probably not representative". This is a real system with real NFM features that were placed there to address a real flood risk issue (The village downstream, Flimby, is categorized as a community at risk by the Environment agency). How can that not be "representative" when it is a real case? Of course, it will not be representative of all other NFM systems, and we are not claiming that it is. We hope that other researchers will consider downloading the model code for applications to other systems elsewhere.

- Resilience of a network of NFM-measures. Not fulfilled. (Resilience is neither defined nor addressed in the results.)

Again, network resilience analysis *per se* is not one of stated objectives. Rather, it can be considered as a potential derived outcome of our objectives (1) and (2c). We have illustrated how system resilience can be addressed in the results (Figs 8 and 10 again) where we examine survival of barriers under an ensemble of loading conditions. We agree that resilience was not defined - proposing to tighten this up.

**Tables of responses to REV 1 and REV 2**

| ID | REV 1 Comment | Response | Proposed change |
|---|---|---|---|
| | **Check of deliveries with issue:** | | |
| 1 | Design advice. Not fulfilled. (No quantification, the given design proposal is not derived from the network model) | We are worried that the first reviewer thinks we were trying to present the method as a "for-real" real risk assessment, rather than as a proof of concept. We were demonstrating the approach as a proof of concept to encourage use of the approach in other configurations and to demonstrate the flexibility to experiment with configurations. | We propose amending the abstract (line 24) and discussion in text on page 3 at lines 19-20 to make it clear that we are demonstrating a model that can be used to help inform the robust design of **networks** of NFM assets (rather than advice about the construction of an individual asset), in the form of "rules of thumb" as discussed in Sections 4 and 5. We wish to change the last sentence of the abstract to: *"The efficient scheme permits rapid assessment of the whole system performance of dams placed in different locations in real networks, demonstrated in application to a real system of leaky barriers built in Penny Gill, a stream in the West Cumbria region of Britain"* |
| 2 | - Understanding of effective risk reduction strategies. Partially fulfilled. (advices are given, but on a predictable level) | We made comments we think are supported by the analysis in terms of width of site, slope of site, location of barriers in network.

 We have avoided making more generic statements without further research and justification – for instance the fragility assumption really needs to be based on observed failure rates with estimated storm probability to take the conclusions further. We have recommended this as an important area for further research. | We will focus on highlighting more results that are unpredictable. We propose adding the following sentence below Figure 14 stating: *"Whilst storage may be improved well above that for the real system ($355m^3$ as opposed to $235 \ m^3$), the dynamic utilisation of that storage in the network does not result in a better reduction in peak flow (97% as opposed to 96%), this being a key measure to assess the effectiveness of the whole system. This highlights the unpredictability of the network and whole system performance and demonstrates why such a model is important at larger scales."* |
| | | | |

|   | **Check of scientific questions:** | | |
|---|---|---|---|
| 3 | Effective dynamic utilization of storage at network dam placements. Partially fulfilled. (There is no criteria given how effectiveness is defined in this context. The effectiveness is not exactly quantified. The effect for the given examples is low. The effectiveness is not linked to the given design flood. | The effectiveness of the dynamic utilisation is expressed in terms of the overall reduction in peak flow whilst considering the number of barriers used. We demonstrate this in the second application to Penny Gill. The important effect here is that whilst we may be able to add lots of storage, it does not always fill and release fast enough that it can be used to take water away from the peak of a storm event.

Conversely there is evidence of some leaky barriers not filling, which also make them less effective. We believe that the different trade-offs in terms of slope, positioning and storage lead to complexity and it is not always possible to pre-determine optimal arrangements, but it is possible to explore with this rapid network model. | Please see our response to last point and proposed amendment.

We propose adding a further comment in the discussion on page 18, line 20: "*The network analysis has also demonstrated how the effectiveness may involve assessing several factors (total potential additional storage volume, maximum dynamic utilisation of that storage, amount of un-used storage volume) that will vary across the system depending on the spatial and temporal pattern of runoff inputs*." |
| 4 | Identification of placements that reduce the risk of (cascade) failure. Partially fulfilled. (just qualitative analysis possible, because both dam leakage and fragility are not known.) | Figures 8-10 and discussion in Section 3.3 provide quantitative analysis of failure risk, and enable a comparison of two idealised network designs. Both leakage and fragility can be parameterised within the model, i.e. treated quantitatively. | We think we have demonstrated that the network model is capable of quantitative analysis of single and cascade failures. |
| 5 | - Do small-scale interventions using NFM (Natural Flood | We demonstrate the model as a proof of concept at the small scale and think this objective is | We propose to make the sentence starting on line 22 more emphatic. Rather than stating " *Probably more* |

| | | | |
|---|---|---|---|
| | Management) combine to create a large scale benefit at large scale? Not fulfilled. (No definition what a large benefit in a large scale is. No model investigation carried out analyzing this particular question. The models used are by far too small.) | the next natural progression of this work. We also identify on Page 3, line 2 how scaling up is a 'significant outstanding research question'. We have not set out to tackle this problem here, but we think simplified network model will help with this. | *importantly….*" To "*However, such general advice can over-simplify, and the final example has demonstrated that this type of model can be effectively used to rapidly test different arrangements of dams and to assess which are likely to work best to reduced risk given the unpredictability of the whole system response*". |
| 6 | - Reliability and performance of NFM-measures under plausible hazards. Not fulfilled. (No plausible hazard given. The test case with Qmax= ca. 10m3/s is completely synthetic. The case study area is not given, but the small N100 resulting in Qmax = 0,5m3/s indicate a very small catchment area, which probably is not representative.) | I think the conclusions for Penny Gill can be drawn out to meet this criterion better, but we explain the choice of flows and storage which I think may have been overlooked.

The initial analysis was deliberately a Synthetic case –as part of proof of concept. Please see this as a test of a method that uses dimensionless equations and can be scaled to more realistic flows. The flows selected were deliberately extreme but were selected (combined with additional storage) to stress-test the barriers for failure. see answer to Rev 2) | We will include more context on Page 2, line 10 about the case study, Penny Gill.
We will comment that although it is a small area (<0.5km$^2$), the tested scenarios do create a plausible risk because of interactions with infrastructure further downstream; in this case the capability to attenuate the peak flows for this small sub-catchment is important to avoid backing up at culverts. These impact the community designated 'at risk' by the Environment Agency in the downstream village of Flimby. |
| 7 | - Resilience of a network of NFM-measures. Not fulfilled. (Resilience is neither defined nor addressed in the results.) | We demonstrate resilience in a number of ways. With regards to the synthetic case it can be interpreted as the survival of the leaky barriers in different configurations and the capacity of the system to deliver a reduction in peak flow within an ensemble of potential forcing events and failure scenarios. We have use graphical | We will reinforce the notion of 'system resilience' by discussing this at Page 8, line 15 in terms of whether the system can still deliver a reduction in flood peak downstream even if elements within it are allowed to fail. This will be discussed in relation to Figure 8, which we have chosen to use in place of a single number. |

| | | representation of resilience in the form of Figure 8. | |
|---|---|---|---|
| 8 | - Definitions on effectiveness, resilience and "large benefit at large scales" | Ok – | We will ensure the terms are defined (for resilience see last answer) such that they can be interpreted using Figs 8 and 10. |
| 9 | - Hydrological aspects (f.i. definition of design hydrograph, impact of convective or orographic rainfall events on efficiency) | ok | We will add a definition of these terms |
| 10 | - Ecological aspects | We didn't set out to cover this but we could add to discussion – the design advice could include fish passage | We will add a discussion of the potential issue of blocking fish passage if the under-flows are too narrow (this was prepared in a previous draft but omitted to save space) |
| 11 | - Arguments for the development of the network model (there are well-proven 1d-tools as MIKE11 or HECRAS) | The packages mentioned do not allow for rapid assessment of collapse and cascade collapse of structures with leakiness factors in arbitrary network configurations.

It would be a lot of work to manually implement the automated solver used here that continues after barrier failure and multiple failures. | We propose clarifying why there are advantages to developing a new model and the ability of the solver to continue following failure or cascade failure by adding this sentence on Page 8, line 38:
*"Whilst there are a number of hydraulic modelling packages solving similar equations with a diverse range of hydraulic units, these do not permit rapid assessment of collapse and cascade collapse of barriers having leakiness factors and a channel storage multiplier making it easier to test arbitrary networks of configurations."* |
| 12 | - Validation of proposed network model for normal flow conditions and dam-break | The case studies are a form of sensitivity analysis – but we agree validation in future useful – good to make comparisons where upstream and downstream measurements – | We will clarify on Page 14, line 9:
*"We consider the application of the model in a 'sensitivity to change investigation' to a site on Penny Gill, West Cumbria (figure 1)."* |

**Response to REV 2 overleaf**

**REVIEWER #2**

| ID | REV #2 Comment | Response | Proposed change |
|----|----------------|----------|-----------------|
| 1 | I really enjoyed this paper. It was novel, innovative and thought provoking. I do have a number of fundamental detailed points that I will outline below. | Thank you | |
| 2 | Overall I am concerned that the RAF/Flood features being shown are not really good features to build or simulate. However, the theoretical analysis suggested that small networks of barriers can have a huge impact on flood reduction. | That is for the hypothetical case investigated in the workshop. We then apply to a more real-world situation | See responses below |
| 3 | This needs a number of clarification points. Firstly the feature shown are small and create small volumes of temporary storage. The features are trapped within the channel and hence seem to have little capacity to store flow. Hence the conclusion must be that it is the roughness that is slowing the flow???? This type of feature works better when the water is forced onto a floodplain and into extra storage areas. So I would be | Please see further responses on the synthetic case, where the λ factor accounts for floodplain storage, and for Penny Gill where the barriers are robust and quite tall.

It is the temporary (in-channel) storage behind the relatively large barriers which is attenuating the peaks. | We will highlight the nature of the theoretical channel, with additional storage represented using the λ factor, and the real channel at Penny Gill.

We will add context at page 14, line 17, that the Penny Gill stream is incised and there is little possibility of greater connection with the floodplain, so hence the barriers are relatively tall and rely on extended in-channel storage. There is one area where there is a wider floodplain that does show potential (see Figure 11, 325m). |

| | | | |
|---|---|---|---|
| | suggesting that RAF design is key, i.e. using the barriers in combination with other storage and velocity reducing zones, e.g. shallow flow across floodplains. | | |
| 4 | A criticism of the Metcalf work is that the features did not allow flow onto to the farmland next to the channels. | This is not a comment on our paper | |
| 5 | However, let's go with the network of within channel features and discuss that. | ok | |
| 6 | One more 'picky' thing the author uses the term 'we' a lot. In the past this would not be allowed but I know we live in enlightened times and we can now use 'we'. I do think we have an overzealous use of the term 'we', especially when occurs 4 or 5 times in paragraph. 'We have formulated a network method…' could be 'A network method has been formulated…' | Ok | We'll take guidance from the editor – happy to change language. |
| 7 | I think earlier work on network models could be included and referenced. E.g Nicholson et al., 2019, Quinn et al., 2015 which was included in the WWNP report. I think Bhoko et al., | Ok. We have found it hard to find research on the performance of the type of leaky barriers under investigation (i.e. solid hose-jump type barriers). | We will add the proposed citations and relate to more current work. |

| | | | |
|---|---|---|---|
| | 219/20? Could be added.
The work of Nicholson included observations and this study has used a theoretical approach and this may need to be highlighted. | | |
| 8 | Bhoko comments on beavers and the impact on flood flow. It is also important to stress that beaver dams are full of water all the time. So that is why they are useless to flooding! | Ok. | To add comment – and reference cited paper. |
| 9 | I think your model is not simulating a typical leaky dam, even though figure 1 is a little leaky. Most leaky dams are usually large woody debris that are very porous/leaky. You are thus assuming that all leaky dams operate as a sluice gate. This may not be a good assumption. But, let's go with the impact of a network of sluice gates on flood flow. | We chose this type of barrier and have seen quite a few in operation (as in the case-study) where the barriers are solid and there is an underflow.

Our model does also include a porosity term. Whilst we set porosity to zero for the cases investigated in the paper, in general this can be changed using the factor k in eq 7. For Penny Gill the leakage is in practice very small compared to the underflow. | To broaden the description of leaky barriers – and reflect the diversity. To emphasise how other types of barrier could be modelled with future adaptation of the equations. Note on Page 18, line 18 (last sentence) we already do emphasise the diversity. |
| 10 | Fig 2 is crucial as is section 2.2. What is the size of the catchment? Add a scale to fig 2. I know you want it to be a dimensionless model, but surely a catchments size would help. You need to state the density of barriers. I deduced later that the model | The synthetic case was developed based on approximate reference scales (the equations are non-dimensionalised). We wanted to stress-test the system using extreme flows that would trigger failures with the assumed fragility function, so that we could demonstrate an approach that can test system resilience.

Please see our response to the next point which should address your concerns here; we think there is a | Please see comments below regarding the use of a large λ factor, which is a convenient way of representing extra channel or floodplain storage.

We propose changing the figure 2 caption to: 'The lengths of the network edges can be varied depending on the scale of interest; in the examples shown in this paper |

| | | | |
|---|---|---|---|
| | had 100m lengths between nodes. Later you show the model resulting as being based on Fig 2b, so this means the length of channel is 500m, This means the catchment is about 1km2? SO, how do you get a flow of 15 m3/s (fig 7). This would be a flow rate for a 10-20km2 catchment. That catchment area would require 100 RAFs. Can the authors sort this out or address my misunderstanding. | potential for misunderstanding that we can avoid with a minor revision. | they are typically on the order of 100m long.' |
| 11 | The model result then suggest a huge reduction in flow caused by 5 barriers?? 15m3/s down to 6m3/s. But the storage behind these barriers must be tiny. Surely they would be overwhelmed by such high flows. SO I may have missed something. SO what is causing the reduction? Is it roughness? Could the authors show some sensitivity analysis of Mannings 'n' on the model output? | There are two factors at work here – 1) we wanted to use extreme flows to test performance, as noted above, and 2) we introduced a channel storage factor, λ, which can represent extra channel or floodplain storage per segment, which as you correctly point out is essential for gaining significant attenuation when using leaky barriers. The parameter λ is introduced in equation (14), which increases the lateral storage or, in effect, represents a well-connected floodplain for values > 1. This enabled us to represent significant storage by setting the value of λ to 50 (see Figure 5) – which assumes **a lot** of extra storage (conceptually a wide floodplain) – and hence perhaps why you were uncomfortable with the magnitude of the reduction. Thus, in these equations we are able to represent more than simply the wedge-storage in the channel behind | We will discuss this in more detail with reference to Figure 5 and where we use extreme flows and the large λ factor. When we use the real case of Penny Gill then the λ factor is 1, which ties in with peak flows and storage volumes that are realistic for this system. |

| | | a barrier. This means we can represent the type of extended floodplain storage that the reviewer has mentioned in previous comment | |
|---|---|---|---|
| 12 | I like the analysis introducing branches, but again I do worry about scale and flows. The point being made about branches have less danger is good, as this reflects that a scale appropriate positioning of RAFs is needed, as they work better and have less problems in smaller channels. | We have addressed this in our response to the previous comment.

The equations are non-dimensional so different assumptions can be made about the appropriate scale. | |
| 13 | Overall I was less impressed with the analyses and cascade failures. I am not addressing the analysis which is good, but a NFM team would either design in the failure so that lower dams trap debris or if there is a threat than a debris traps would be built. | In terms of design taking in these considerations there is no current definitive guidance in the UK, (although CIRIA are in the process) so 'NFM teams' do need some guidance, and internationally, The WWF "Green Atlas" (https://www.worldwildlife.org/publications/natural-and-nature-based-flood-management-a-green-guide) provides case studies to advise on 'good practice' | To discuss trapping of failed barriers more – although we do already state that the Penny Gill implementation has mitigated this to some extent (see last sentence of page 18). |
| 14 | SO, now we look at Penny Gill, the flow rate is 0.4m3/s. This analysis section is interesting. However, it reinforces my point about the flow rates used above. You really have a chalk and cheese comparison. The features in Penny Gill should have been better designed, i.e. to use zones with active floodplains (if | We were using Penny Gill to illustrate a simple application – not purpose of paper to debate whether it was the most appropriate choice of NFM system design.

Agree the scales are different – we were partially addressing the problem that you had with the synthetic workshop example, and wanted to apply the proof of concept derived in a workshop to a 'real world case' | See responses above (point 6 for first reviewer and point 3 for the second reviewer). |

| | | | |
|---|---|---|---|
| | appropriate) or maybe the wood should have been distributed over larger areas to create more roughness, thus could be more tree planting in the channel, e.g. willow. | The incision in the Gill makes what you state difficult – there is one area where the floodplain is within reach of realistic flood flows and that was the one wide area identified in Figure 11, 325m.

The West Cumbria Rivers Trust have sought as much wedge-storage as possible in the gill, but we do advise like you to reconnect floodplain where possible. However as stated the channel is highly incised. | |
| 15 | I hope the authors do not think I am being over critical. I really did enjoy the paper. I may not fully understand the model, the scale, the flows, the volumes and the RAF density assumptions. If you sort this out then this paper can be published. If it is a simple clarification of the model and the assumptions then this would be just a minor edit. If there is a fundamental issue with the flow and the impact then major corrections are needed. | No - these are good points, thank you, and help improve the paper.

In the synthetic example from the workshop, we chose to scale the non-dimensional equations with values that yielded interesting cases – such as failure and attenuation.

We could have chosen small values of $\lambda$ to reflect many installations of leaky barriers we have seen, but there would have been little response in the hydrograph and this would reflect your comments on the need to reconnect floodplain storage. | Hopefully the various clarifications above and our introductory remarks address these concerns. |

---

## Author Comment (AC2) · 6 Jul 2020

Please see attached letter and author response to comments, Thank you

Please also note the supplement to this comment:
https://www.nat-hazards-earth-syst-sci-discuss.net/nhess-2019-394/nhess-2019-394-AC2-supplement.pdf

---

## Author Response (AR1)

**Author Comments (version 2) to review of "Network performance of leaky barriers NHESS 2019-394 Discussion Paper".** B. Hankin, R. Lamb, I. Hewitt, 14-07-2020.

Dear Editor and Reviewers,

Thank you very much for the review comments on our NHESS discussion paper on a network performance model for leaky barriers. We have provided a response to each reviewer comment in the table below and think the proposed changes and corrections will lead to an improved and clearer manuscript.

This paper came out of an applied mathematics group challenge to understand the performance of a network of nature-based, leaky-barriers and to establish a fast model capable for testing many configurations at larger scales as a proof-of-concept. Our preliminary investigation uses a synthetic case with *extreme* flows, specifically chosen to stress-test the network and produce failures (and cascade failures) of the leaky barriers. These test cases also *incorporate substantial storage* (which may not be apparent from a first look at the equations), made possible using a convenient factor, $\lambda$, which may have been overlooked in the review. We are very much in agreement regarding the need for leaky-barrier type solutions to connect to as much storage as possible and the formulation provided with a solution in one-dimensional format permits such testing for a wide range of situations.

We first use the synthetic case to establish that the model can account for open channel flow with variable storage in 2d networks of leaky barriers whilst incorporating under-flow, over-flow, porous through-flow, barrier failure and cascade failure. We demonstrate its flexibility in application to a real network of existing barriers, and consider that the paper and model we have provided on *gitlab* will help others to answer pressing questions on configuring spatial strategies for large networks of leaky barriers.

In addition, in response to editor comments, we have undertaken the following:

(a) Added a legend to Figures 8 and 10, so it is obvious what the size and colour mean without searching in the text.

(b) Removed footnotes.

(c) Corrected numbering of 'Code availability' and 'References'.

**We first make some general comments in response to Reviewer #1 and then tabulate our response to each reviewer point in the table further below**

**Reviewer #1**

We have assessed the comments made by Reviewer #1. We have looked to address the constructive suggestions for improvements in the reviewer's comments; our proposed responses are given in a table that follows. We also have some general comments in response to this review, where we denote text taken from the reviewer's commentary in red.

The reviewer structured their comments around a cross-check of "Deliveries and scientific questions as defined by the author Barry Hankin."

We believe that in formulating their "check list" the reviewer has gone beyond the actual objectives we stated, as we explain below. Whilst we would concur that the objectives suggested by the reviewer are all of interest, they do not altogether represent our objectives in this paper.

**REVIEWER #1 Check of deliveries:**

- Delivery of network based model. Fulfilled. (215 lines matlab- code; 25 lines calculation, 190 lines I/O and plotting)

Agreed.

- Rapid assessment. Fulfilled.

Agreed.

- Design advice. Not fulfilled. (No quantification, the given design proposal is not derived from the network model)

We did not set out to offer design advice, believing that this would be premature. We are not aware of having set out any specific "given design proposal". Rather, our hope is that the paper demonstrates a method that could be applied further to inform design advice with respect to the configuration of systems of leaky barriers (as opposed to construction advice about individual assets).

We think the wording of our original text in places does not reflect this position clearly enough and propose to amend it accordingly. The first sentence in the Conclusions sums up our position "We have formulated a network model for a catchment area that **allows for** simple exploration of the effectiveness of different dam placements and designs and is **sufficiently cheap to solve that it may be useful** in analysing risks that require a large ensemble of simulations." (with emphasis added here to stress that we see this as enabling the application to inform design advice, rather than delivering design advice *per se*).

- Understanding of effective risk reduction strategies. Partially fulfilled. (advices are given, but on a predictable level)

This was not one of our stated objectives as written in general terms by the reviewer. Although it can be inferred that the analysis of NFM systems does, overall, seek to understand effective risk reduction strategies, our objectives stated on page 3 were in fact more specific, namely:

1) We want to understand NFM features as systems of assets, and to assess those systems within a risk-based analysis that considers the whole-system performance in terms of risk reduction
2) We wish to understand the impact of different spatial configurations of the leaky barriers, taking into consideration three possible performance issues:
    a. under-utilisation / redundancy
    b. undesired synchronisation
    c. structure failures including cascades

**Check of scientific questions:**

- Effective dynamic utilization of storage at network dam placements. Partially fulfilled. (There is no criteria given how effectiveness is defined in this context. The effectiveness is not exactly quantified. The effect for the given examples is low. The effectiveness is not linked to the given design flood.)

This is one of stated objectives (2a above). We agree it has been fulfilled in part and have made amendments to clarify how we define effectiveness (please see table and definitions in text based on percentage peak flow reduction).

- Identification of placements that reduce the risk of (cascade) failure. Partially fulfilled. (just qualitative analysis possible, because both dam leakage and fragility are not known.)

This is not one of our stated objectives. Our objective was to understand the impact of different spatial configurations of barrier networks, including the possibility of single and cascade failures (2c, above). This could of course help to identify placements that reduce the risk, as we have discussed; however, we did not set out to identify specific placements (in an optimisation sense) as one of our objectives, only broad strategies.

We disagree with the reviewer's assertion that the analysis is qualitative. The results are very clearly quantified in the graphs plotted in Figures 8 and 10. Dam leakage and fragility are explicitly parameterised (i.e. quantitatively) in our model. The selection of suitable values for the leakage and fragility functions is of course a different question; in the discussion and conclusions we comment on the desirability of calibration in future applications of the model.

- Do small-scale interventions using NFM (Natural Flood Management) combine to create a large scale benefit at large scale? Not fulfilled. (No definition what a large benefit in a large scale is. No model investigation carried out analysing this particular question. The models used are by far too small.)

This was not one of our stated objectives.

It is an important, broader question that we raised for context in the introduction to the paper. We also comment in our discussion and conclusion that our model could be helpful in exploring this question further. But we did not set out to tackle scaling in this paper. Rather, we see the paper as a precursor step to demonstrate a model that is capable of scaling, which could be applied further as we mention in the discussion and conclusions.

- Reliability and performance of NFM-measures under plausible hazards. Not fulfilled. (No plausible hazard given. The test case with Qmax= ca. 10m3/s is completely synthetic. The case study area is not given, but the small N100 resulting in Qmax = 0,5m3/s indicate a very small catchment area, which probably is not representative.)

We agree that the plausibility of the test cases require further discussion and clarification. We have given details of proposed amendments in our response to Reviewer #2, who raised some specific questions about this issue. Reviewer #2 identified some potential for misunderstanding about how additional channel/floodplain storage is represented in the equations, which we propose to clarify through amendments (see table).

We cannot agree with the comment that the small catchment area in Penny Gill is "probably not representative". This is a real system with real NFM features that were placed there to address a real flood risk issue (The village downstream, Flimby, is categorized as a community at risk by the Environment Agency – flooding is caused by backing up upstream of culverts where any reduction in peak flow would be useful). How can that not be "representative" when it is a real case? Of course, it will not be representative of all other NFM systems, and we are not claiming that it is. We hope that other researchers will consider downloading the model code for applications to other systems elsewhere.

- Resilience of a network of NFM-measures. Not fulfilled. (Resilience is neither defined nor addressed in the results.)

Again, network resilience analysis *per se* is not one of stated objectives. Rather, it can be considered as a potential derived outcome of our objectives (1) and (2c). We have illustrated how system resilience can be addressed in the results (Figs 8 and 10 again) where we examine survival of barriers under an ensemble of loading conditions. We agree that resilience was not defined – and have tightened this up – see table below.

**Tables of responses to REV 1 and REV 2**

| ID | REV 1 Comment | Response | Proposed change |
|---|---|---|---|
| | **Check of deliveries with issue:** | | |
| 1 | Design advice. Not fulfilled. (No quantification, the given design proposal is not derived from the network model) | We are worried that the first reviewer thinks we were trying to present the method as a "for-real" real risk assessment, rather than as a proof of concept. We were demonstrating the approach as a proof of concept to encourage use of the approach in other configurations and to demonstrate the flexibility to experiment with configurations. | We have amended the abstract (line 24) and discussion in text on page 3 at lines 19-20 to make it clear that we are demonstrating a model that can be used to help inform the robust design of **networks** of NFM assets (rather than advice about the construction of an individual asset). We have changed the last sentence of the abstract to: *"The efficient scheme permits rapid assessment of the whole system performance of dams placed in different locations in real networks, demonstrated in application to a real system of leaky barriers built in Penny Gill, a stream in the West Cumbria region of Britain"* |
| 2 | - Understanding of effective risk reduction strategies. Partially fulfilled. (advices are given, but on a predictable level) | We made comments we think are supported by the analysis in terms of width of site, slope of site, location of barriers in network.

We have avoided making more generic statements without further research and justification – for instance, the fragility assumption really needs to be based on observed failure rates with estimated storm probability to take the conclusions further. We have recommended this as an important area for further research. | We have definitions of effectiveness in Section 1.0 (final paragraph) and added the following sentence below Figure 14 quantifying the system effectiveness: *"Whilst storage may be improved well above that for the real system (355m$^3$ as opposed to 235 m$^3$), the dynamic utilisation of that storage in the network does not result in a better reduction in peak flow (97% as opposed to 96%), this being a key measure to assess the effectiveness of the whole system. This highlights the unpredictability of the network and whole system performance and demonstrates why such a model is important at larger scales."* |
| | | | |

| | Check of scientific questions: | | |
|---|---|---|---|
| 3 | Effective dynamic utilization of storage at network dam placements. Partially fulfilled. (There is no criteria given how effectiveness is defined in this context. The effectiveness is not exactly quantified. The effect for the given examples is low. The effectiveness is not linked to the given design flood. | The effectiveness of the dynamic utilisation is expressed in terms of the overall reduction in peak flow whilst considering the number of barriers used. We demonstrate this in the second application to Penny Gill. The important effect here is that whilst we may be able to add lots of storage, it does not always fill and release fast enough that it can be used to take water away from the peak of a storm event.

Conversely there is evidence of some leaky barriers not filling, which also make them less effective. We believe that the different trade-offs in terms of slope, positioning and storage lead to complexity and it is not always possible to pre-determine optimal arrangements, but it is possible to explore with this rapid network model. | Please see our response to last point and amendment.

We have added a further comment in the discussion in the last para of Section 4:

*"In summary, the network analysis has demonstrated how the effectiveness of the system of leaky-barriers quantified overall using the integrating measure of percentage peak flow reduction at the bottom of the network. The approach accounts for additional storage volumes put in place, but also how it is utilised dynamically, something that will also vary across the system depending on the spatial and temporal pattern of runoff inputs, slot dimensions and leakiness."* |
| 4 | Identification of placements that reduce the risk of (cascade) failure. Partially fulfilled. (just qualitative analysis possible, because both dam leakage and fragility are not known.) | Figures 8-10 and discussion in Section 3.3 provide quantitative analysis of failure risk, and enable a comparison of two idealised network designs. Both leakage and fragility can be parameterised within the model, i.e. treated quantitatively. | We have demonstrated that the network model is capable of quantitative analysis of single and cascade failures graphically using Figures 8 / 10 and through the discussions of percentage change to peak flow. |
| 5 | - Do small-scale interventions using NFM (Natural Flood Management) | We demonstrate the model as a proof of concept at the small scale and think this objective is the next natural progression of | We have made the sentence starting on line 22 more emphatic in relation to the need for a network model. |

| | | | |
|---|---|---|---|
| | combine to create a large scale benefit at large scale? Not fulfilled. (No definition what a large benefit in a large scale is. No model investigation carried out analyzing this particular question. The models used are by far too small.) | this work. We also identify on Page 3, line 2 how scaling up is a 'significant outstanding research question'. We have not set out to tackle this problem here, but we think simplified network model will help with this. | Rather than stating *" Probably more importantly…."* To: *"However, general advice on design can over-simplify, and the final example has demonstrated that this type of network model can be used effectively to rapidly test different arrangements of dams and to assess which are likely to work best to reduced risk given the unpredictability of the whole system response".* Please also see our discussion of scaling and future research that was already in the last paragraph of the paper. |
| 6 | - Reliability and performance of NFM-measures under plausible hazards. Not fulfilled. (No plausible hazard given. The test case with Qmax= ca. 10m3/s is completely synthetic. The case study area is not given, but the small N100 resulting in Qmax = 0,5m3/s indicate a very small catchment area, which probably is not representative.) | We explain the choice of flows and storage which we think may have been overlooked. The initial analysis was deliberately a synthetic case –as part of proof of concept. Please see this as a test of a method that uses dimensionless equations and can be scaled to other flows. The synthetic flows selected were deliberately extreme but were selected (combined with additional storage) to stress-test the barriers for failure. see answer to Rev 2). The flows for Penny Gill are realistic and due to backing up at infrastructure, flooding does occur. | We have included more context on Page 2, line 10 about the case study, Penny Gill. The text now reads: *Penny Gill micro-catchment, drains to the small community at risk of Flimby on the west coast of Cumbria and is designated at risk because of the interaction of the stream with infrastructure downstream of the test-site. In this case the capability to attenuate the peak flows for this small sub-catchment ($<0.5km^2$) is important to avoid backing up and flooding from culverts.* |
| 7 | - Resilience of a network of NFM-measures. Not fulfilled. (Resilience is neither defined nor addressed in the results.) | We demonstrate resilience in a number of ways. With regards to the synthetic case it can be interpreted as the survival of the leaky barriers in different configurations and the capacity of the system to deliver a reduction in peak flow within an ensemble of potential forcing | We reinforce the notion of 'system resilience' by discussing this at Page 12, line 15 and at the bottom of original page 13 in terms of whether the system can still deliver a reduction in flood peak downstream even if elements within it are allowed to fail. This has now been discussed more fully in relation to Figure 8/10, which |

| | | events and failure scenarios. We have use graphical representation of resilience in the form of Figure 8. | we have chosen to use in place of a single number. |
|---|---|---|---|
| 8 | - Definitions on effectiveness, resilience and "large benefit at large scales" | Ok – | We have ensured these terms are defined (for resilience see last answer) such that they can be interpreted using Figs 8 and 10. See page 2, line 2 and last paragraph of Section 1.0 where we have specified the measures of percentage peak flow reduction and frequency of barrier failure. See paragraph under figure 14. |
| 9 | - Hydrological aspects (f.i. definition of design hydrograph, impact of convective or orographic rainfall events on efficiency) | ok | See definition of 'design event' in first paragraph of section 4.0 and definitions of effectiveness discussed above. |
| 10 | - Ecological aspects | We didn't set out to cover this but we could add to discussion – the design advice could include fish passage | We have referenced the potential issue of blocking fish passage if the under-flows are too narrow (this was prepared in a previous draft but omitted to save space). Text added P18: "*These different designs also need to consider the trade-offs in barrier design (slot height, leakiness) when considering ecological impacts such as fish passage, for example a narrow slot might improve flood attenuation but make passage more difficult*" |
| 11 | - Arguments for the development of the network model (there are well-proven 1d-tools as MIKE11 or HECRAS) | The packages mentioned do not allow for rapid assessment of collapse and cascade collapse of structures with leakiness factors in arbitrary network configurations.

It would be a lot of work to manually implement the automated solver used here that *continues* after barrier failure and multiple failures. | We have clarified why there are advantages to developing a new model and the ability of the solver to continue following failure or cascade failure by adding this sentence on Page 8, line 38: "*Whilst there are a number of hydraulic modelling packages solving similar equations with a diverse range of hydraulic units, these do not permit rapid assessment of collapse and cascade collapse of barriers having leakiness factors and a channel* |

| | | | storage multiplier making it easier to test arbitrary networks of configurations." |
|---|---|---|---|
| 12 | - Validation of proposed network model for normal flow conditions and dam-break | The case studies are a form of sensitivity analysis – but we agree validation in future useful – it would be good to make comparisons where upstream and downstream measurements are available | We clarify on Page 14, line 9: *"We consider the application of the model in a 'sensitivity to change investigation' to a site on Penny Gill, West Cumbria (figure 1)."* |

**Response to REV 2 overleaf**

**REVIEWER #2**

| ID | REV #2 Comment | Response | Proposed change |
|----|----------------|----------|-----------------|
| 1 | I really enjoyed this paper. It was novel, innovative and thought provoking. I do have a number of fundamental detailed points that I will outline below. | Thank you | |
| 2 | Overall I am concerned that the RAF/Flood features being shown are not really good features to build or simulate. However, the theoretical analysis suggested that small networks of barriers can have a huge impact on flood reduction. | That is for the hypothetical case investigated in the workshop. We then apply to a more real-world situation that was already constructed. | See responses below with regards to an error in figure 5, but especially in relation to the λ factor which permits investigation of more storage including floodplain storage. |
| 3 | This needs a number of clarification points. Firstly the feature shown are small and create small volumes of temporary storage. The features are trapped within the channel and hence seem to have little capacity to store flow. Hence the conclusion must be that it is the roughness that is slowing the flow???? This type of feature works better when the water is forced onto a floodplain and into extra storage areas. So I would be | Please see further responses on the synthetic case, where the λ factor accounts for floodplain storage, and for Penny Gill where the barriers are robust and quite tall.

It is the temporary (in-channel) storage behind the relatively large barriers which is attenuating the peaks, combined with the roughness.

This system has been built and is not at the design stage. | We have highlighted the nature of the theoretical channel, with additional storage represented using the λ factor, and for the real channel at Penny Gill we have added more context under figure 11:

*Penny Gill stream is incised and there is little possibility of greater connection with the floodplain, so hence the barriers are relatively tall and rely on extended in-channel storage, except for the relatively wide segment in figure 11 which reflects an area of channel well* |

| | | | |
|---|---|---|---|
| | suggesting that RAF design is key, i.e. using the barriers in combination with other storage and velocity reducing zones, e.g. shallow flow across floodplains. | | *connected with a relatively wide depression.* |
| 4 | A criticism of the Metcalf work is that the features did not allow flow onto to the farmland next to the channels. | This is not a comment on our paper | |
| 5 | However, let's go with the network of within channel features and discuss that. | ok | |
| 6 | One more 'picky' thing the author uses the term 'we' a lot. In the past this would not be allowed but I know we live in enlightened times and we can now use 'we'. I do think we have an overzealous use of the term 'we', especially when occurs 4 or 5 times in paragraph. 'We have formulated a network method…' could be 'A network method has been formulated…' | Ok | The editor has stated he does not have a language preference in this respect. |
| 7 | I think earlier work on network models could be included and referenced. E.g Nicholson et al., 2019, Quinn et al., 2015 which was included in the WWNP report. I think Bhoko et al., | Ok. We have found it hard to find research on the performance of the type of leaky barriers under investigation (i.e. solid hose-jump type barriers). | The Nicholson et al paper is already cited on page 8 line 16 of original submission and is relevant. Quinn reference added in same place based on the WWNP citation (2013) |

| | | | |
|---|---|---|---|
| | 219/20? Could be added.
The work of Nicholson included observations and this study has used a theoretical approach and this may need to be highlighted. | | |
| 8 | Bhoko comments on beavers and the impact on flood flow. It is also important to stress that beaver dams are full of water all the time. So that is why they are useless to flooding! | Ok. | This has not been included since the main point we wished to make here in relation to beaver dams is to learn about performance failure; in this case we assume that leaky-barriers for NFM are designed not to be full of water before a flood (ie they are leaky with additional storage that is used at high flows) |
| 9 | I think your model is not simulating a typical leaky dam, even though figure 1 is a little leaky. Most leaky dams are usually large woody debris that are very porous/leaky. You are thus assuming that all leaky dams operate as a sluice gate. This may not be a good assumption. But, let's go with the impact of a network of sluice gates on flood flow. | We chose this type of barrier and have seen quite a few in operation (as in the case-study) where the barriers are solid and there is an underflow.

Our model does also include a porosity term. Whilst we set porosity to zero for the cases investigated in the paper, in general this can be changed using the factor k in eq 7. For Penny Gill the leakage is in practice very small compared to the underflow. | Note on Page 18, line 18 (last sentence) we already do emphasise the diversity in leaky barriers:

"..*compounding factor is that there are many forms of leaky barrier or large woody debris dams (Addy et al., 2019) including placing large woody debris in channel, to the horse-jump type barriers in use in Penny Gill, combined with engineered log jams which will also reduce passage of debris should a structure fail.*"

In this location we now also note that the leakiness factor in the equations, in addition to the slot dimensions can be experimented with to represent a broader range of features. |
| 10 | Fig 2 is crucial as is section 2.2. What is the size of the | The synthetic case was developed based on approximate reference scales (the equations are non- | There was a mistake in the left hand size of Figure 5 that has now been amended and |

| | | | |
|---|---|---|---|
| | catchment? Add a scale to fig 2. I know you want it to be a dimensionless model, but surely a catchments size would help. You need to state the density of barriers. I deduced later that the model had 100m lengths between nodes. Later you show the model resulting as being based on Fig 2b, so this means the length of channel is 500m, This means the catchment is about 1km2? SO, how do you get a flow of 15 m3/s (fig 7). This would be a flow rate for a 10-20km2 catchment. That catchment area would require 100 RAFs. Can the authors sort this out or address my misunderstanding. | dimensionalised). We wanted to stress-test the system using extreme flows that would trigger failures with the assumed fragility function, so that we could demonstrate an approach that can test system resilience.

Please see our response to the next point which should address your concerns here; we think there is a potential for misunderstanding that we can avoid with a minor revision. | reported with track changes – the peak flow was 10m³/s not 15m³/s that was from an earlier numerical experiment.

Above equation (15) we have added the text:
*Here an extreme flow of 10 m³/s is used to fully stress-test the system, also permitting flows and depths with magnitudes capable of failing leaky-barriers.*

We have changed figure 2 caption to add the text:
*'The lengths of the network edges can be varied depending on the scale of interest; in the examples shown in this paper they are typically on the order of 100m long.'*

This should be considered in combination with the fact a large $\lambda$ factor has been used so there is adequate storage capable of providing attenuation. |
| 11 | The model result then suggest a huge reduction in flow caused by 5 barriers?? 15m3/s down to 6m3/s. But the storage behind these barriers must be tiny. Surely they would be overwhelmed by such high flows. SO I may have missed something. SO what is causing the reduction? Is it roughness? Could the authors show some | There was a mistake in the legend for figure 5 – see above.

There are also two factors at work here – 1) we wanted to use extreme flows to test performance, as noted above, and 2) we introduced a channel storage factor, $\lambda$, which can represent extra channel or floodplain storage per segment, which as you correctly point out is essential for gaining significant attenuation when using leaky barriers.

The parameter $\lambda$ is introduced in equation (14), which increases the lateral storage or, in effect, represents | Figure 5 has been corrected.

Please see new edit below regarding the use of a large $\lambda$ factor, which is a convenient way of representing extra channel or floodplain storage. This is highlighted under equation 14 in relation to this discussion:

"*where $\lambda \geq 1$ is this enhancement factor that accounts for a larger volume being stored behind the dam. In practice, this would have to be estimated for each dam* |

| | | | |
|---|---|---|---|
| | sensitivity analysis of Mannings 'n' on the model output? | a well-connected floodplain for values > 1.

This enabled us to represent significant storage by setting the value of λ to 50 (see Figure 5) – which assumes **a lot** of extra storage (conceptually a wide floodplain) – and hence perhaps why you were uncomfortable with the magnitude of the reduction.

Thus, in these equations we are able to represent more than simply the wedge-storage in the channel behind a barrier. This means we can represent the type of extended floodplain storage that the reviewer has mentioned in previous comment | *location, but provides a useful mechanism for exploring different NFM designs, for instance it can be used to conceptualise reaches where leaky barriers are being used to enhance floodplain reconnection, thus accessing additional storage with greater potential for peak flow attenuation."*

When we use the real case of Penny Gill then the λ factor is 1, which ties in with peak flows and storage volumes that are realistic for this system. |
| 12 | I like the analysis introducing branches, but again I do worry about scale and flows. The point being made about branches have less danger is good, as this reflects that a scale appropriate positioning of RAFs is needed, as they work better and have less problems in smaller channels. | We have addressed this in our response to the previous comment.

The equations are non-dimensional so different assumptions can be made about the appropriate scale, and we have provided the model. | See discussion in last paragraph of paper about scaling up. |
| 13 | Overall I was less impressed with the analyses and cascade failures. I am not addressing the analysis which is good, but a NFM team would either design in the failure so that lower dams trap debris or if there is a threat than a debris traps would be built. | In terms of design taking in these considerations there is no current definitive guidance in the UK, (although CIRIA are in the process) so 'NFM teams' do need some guidance, and internationally, The WWF "Green Atlas" (https://www.worldwildlife.org/publications/natural-and-nature-based-flood-management-a-green-guide) provides case studies to advise on 'good practice' | We do already state that the Penny Gill implementation has mitigated risk of failure this to some extent through the use of logs within the channel that would trap debris (see penultimate sentence of section 4). |

| 14 | SO, now we look at Penny Gill, the flow rate is 0.4m3/s. This analysis section is interesting. However, it reinforces my point about the flow rates used above. You really have a chalk and cheese comparison. The features in Penny Gill should have been better designed, i.e. to use zones with active floodplains (if appropriate) or maybe the wood should have been distributed over larger areas to create more roughness, thus could be more tree planting in the channel, e.g. willow. | We were using Penny Gill to illustrate a simple application – not purpose of paper to debate whether it was the most appropriate choice of NFM system design. It has already been built and we were studying it.

Agree the scales are different – we were partially addressing the problem that you had with the synthetic workshop example, and wanted to apply the proof of concept derived in a workshop to a 'real world case'

The incision in the Gill makes what you state difficult – there is one area where the floodplain is within reach of realistic flood flows and that was the one wide area identified in Figure 11, 325m.

The West Cumbria Rivers Trust have sought as much wedge-storage as possible in the gill, but we do advise like you to reconnect floodplain where possible. However as stated the channel is highly incised. | See responses above (point 6 for first reviewer and point 3 for the second reviewer).

It is useful to show the flexibility across different situations. |
| 15 | I hope the authors do not think I am being over critical. I really did enjoy the paper. I may not fully understand the model, the scale, the flows, the volumes and the RAF density assumptions. If you sort this out then this paper can be published. If it is a simple clarification of the model and the assumptions then this would be just a minor edit. If there is a fundamental issue | No - these are good points, thank you, and help improve the paper.

In the synthetic example from the workshop, we chose to scale the non-dimensional equations with values that yielded interesting cases – such as failure and attenuation.

We could have chosen small values of λ to reflect many installations of leaky barriers we have seen, but there would have been little response in the hydrograph and this would reflect your comments on the need to reconnect floodplain storage. | Hopefully the various clarifications above, corrections to some of the legends and our introductory remarks address these concerns. |

[revised manuscript text omitted]

---

## Author Response (AR2)

nhess-2019-394

Dear Natascha,

Thank you for accepting our revised NHESS paper.
We would like to thank the Reviewer for accepting the paper as publishable in its current form.
We would like to decline adding the 'optional' additional graph suggested by the Reviewer of total storage versus peak flow reduction. This is partly because there are already a large number of graphs in the paper, but also we have already discussed how there isn't a strong relationship between these two variables in relation to particular barrier configurations. - We think this illustrates why a fast network model is useful to test different storage configurations.

Kind regards,
Barry